# Do We Always Need to Penalize Variance of Losses for Learning with Label Noise?

## Abstract

Algorithms which minimize the averaged loss have been widely designed for dealing with noisy labels. Intuitively, when there is a finite training sample, penalizing the variance of losses will improve the stability and generalization of the algorithms. Interestingly, we found that the variance of losses sometimes needs to be increased for the problem of learning with noisy labels. Specifically, increasing the variance of losses would boost the memorization effect and reduce the harmfulness of incorrect labels. Regularizers can be easily designed to increase the variance of losses and be plugged in many existing algorithms. Empirically, the proposed method by increasing the variance of losses could improve the generalization ability of baselines on both synthetic and real-world datasets.

## 1 Introduction

Learning with noisy labels can be dated back to 1980s (Angluin & Laird, 1988). It has recently drawn a lot of attention (Liu & Tao, 2015; Nguyen et al., 2019; Li et al., 2020; 2021) because large-scale datasets used in training modern deep learning models can easily contain label noise, *e.g.*, ImageNet (Deng et al., 2009) and Clothing1M (Xiao et al., 2015). The reason is that it is expensive and sometimes infeasible to accurately annotate large-scale datasets. Meanwhile, many cheap but imperfect surrogates such as crowdsourcing and web crawling are widely used to build large-scale datasets. Training with such data can lead to poor generalization abilities of modern deep learning models because they will overfit noisy labels (Han et al., 2018b; Zhang et al., 2021).

Generally, the algorithms of learning with noisy labels can be divided into two categories: *statistically inconsistent algorithms* and *statistically consistent algorithms*. Methods in the first category are heuristic, such as selecting reliable examples to train model (Han et al., 2018b; Malach & Shalev-Shwartz, 2017; Ren et al., 2018; Jiang et al., 2018), correcting labels (Ma et al., 2018; Kremer et al., 2018; Tanaka et al., 2018; Reed et al., 2014), and adding regularization (Han et al., 2018a; Guo et al., 2018; Veit et al., 2017; Liu et al., 2020). Those methods empirically perform well. However, it is not guaranteed that the classifiers learned from noisy data are statistically consistent and often need extensive hyper-parameter tuning on clean data.

To address this problem, many researchers explore algorithms in the second category. Those algorithms aim to learn *statically consistent classifiers* (Liu & Tao, 2015; Patrini et al., 2017; Liu et al., 2020; Xia et al., 2020). Specifically, their objective functions are specially designed to ensure that minimizing their expected risks on the noise domain is equivalent to minimizing the expected risk on the clean domain. In practice, it is infeasible to calculate the expected risk. To approximate the expected risk, existing methods minimize the empirical risks, *i.e.*, the averaged loss over the noisy training examples, which is an unbiased estimator to the expected risk (Xia et al., 2019; Li et al., 2021) because their difference will vanish when the training sample size goes to infinity. However, when the number of examples is limited, the variance of the empirical risk could be high, which leads to a large estimation error.

However, we report that penalizing the variance of losses is not always helpful for the problem of learning with noisy labels. By contrast, in most cases, we need to increase the variance of losses, which will boost the memorization effect (Bai & Liu, 2021) and reduce the harmfulness of incorrect labels. This is because deep neural networks tend to learn easy and majority patterns first due to the memorization effect (Bai & Liu, 2021; Zhang et al., 2021). The incorrectly labeled data is of minority and has a more complex relationship between instances and labels compared with correctly

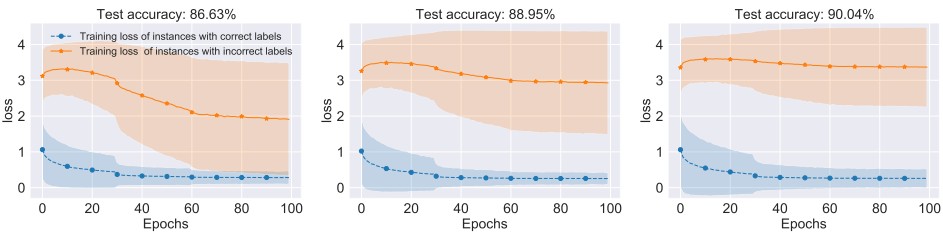

(a) Penalizing variance of losses     (b) Original loss     (c) Increasing variance of losses

Figure 1: We visualize the averaged training loss of instances with correct labels (blue dashed lines) and instances with incorrect labels (yellow solid lines) obtained by penalizing the variance of losses, employing original loss, and increasing the variance of losses in (a)-(c), respectively. The dataset is CIFAR-10 with symmetry-flipping noise, and the noise rate is 0.2. The neural network ResNet-18 and the baseline Forward (Patrini et al., 2017) are employed. The transition matrix $\boldsymbol{T}$ is given and does not need to be estimated.

labeled data, then incorrectly labeled data is harder for neural networks to remember. Therefore, the losses of instances with incorrect labels are likely to be larger than those of correct instances (Han et al., 2018b). Penalizing the variance of losses could force the model to reduce the loss of the instances with incorrect labels because the correct labels are of majority and their losses are smaller, making it hard to distinguish correctly and incorrectly labeled data and will lead to performance degeneration. In contrast, increasing the variance of losses could efficiently prevent large losses from decreasing, then the model may not overfit instances with incorrect labels. In Section 3, we further show that increasing the variance of losses can be seen as a weighting method that assigns small weights to the gradients of large losses and large weights to the gradients of small losses, which could reduce the effect of instances with incorrect labels when updating model's parameters. More discussions about the memorization effect can be found in the Appendix.

Intuitively, as illustrated in Fig. 1, change of the variance of losses does not have much influence on the averaged training loss of instances with correct labels, but makes the averaged training loss of instances with incorrect labels very different. Specifically, penalizing the variance of losses leads to the averaged training loss of instances with incorrect labels decreasing fast, which will encourage the model to overfit instances with incorrect labels. On the contrary, increasing variance of losses can prevent the averaged training loss of instances with incorrect labels from decreasing as shown in Fig. 1c. Therefore, the memorization effect are boosted. As a result, the test accuracy is improved significantly by encouraging the variance of losses.

From the empirical risk minimization perspective, we are encouraged to reduce the variance of losses to increase algorithmic stability. However, to handle label noise, as explained, we may need to boost the variance of losses. This implies that the label noise issue should be carefully considered when designing the loss variance part of learning algorithms. We empirically report that the variance of losses should be boosted in most settings of learning with noisy labels studied in the literature.

The rest of this paper is organized as follows. In Section 2, we introduce related work. In Section 3, we propose our method and its advantages. Experimental results on both synthetic and real-world datasets are provided in Section 4. Finally, we conclude the paper in Section 5.

## 2 RELATED WORK

Some methods proposed to reduce the side-effect of noisy labels using heuristics, For example, many methods utilize the memorization effect to select reliable examples (Han et al., 2020; Yao et al., 2020a; Yu et al., 2019; Jiang et al., 2018) or to correct labels (Ma et al., 2018; Kremer et al., 2018; Tanaka et al., 2018; Reed et al., 2014). Those methods empirically perform well. However, most of them do not provide statistical guarantees for the learned classifiers on noisy data. Some methods treat incorrect labels as outliers and focus on designing bounded loss functions (Ghosh et al., 2017; Gong et al., 2018; Wang et al., 2019; Shu et al., 2020). For example, a symmetric cross-entropy loss has been proposed which has proven to be robust to label noise asymptotically (Wang et al., 2019). These methods focus on the numerical property of loss functions, and the designed loss function can be proved to be noise-tolerant if the noise rate is not large.

The label noise transition matrix $\boldsymbol{T}(x) \in [0, 1]^{C \times C}$ (Patrini et al., 2017; Liu & Tao, 2015; Li et al., 2021), where $C$ is the number of classes, has been widely employed to design statistically consistent

classifiers (Liu & Tao, 2015; Patrini et al., 2017; Xia et al., 2020; Li et al., 2021). Let the clean class posterior $P(\boldsymbol{Y}|X = \boldsymbol{x}) := [P(Y = 1|X = \boldsymbol{x}), \ldots, P(Y = C|X = \boldsymbol{x})]^\top$. It can be inferred by utilizing the noisy class posterior $P(\tilde{\boldsymbol{Y}}|X = \boldsymbol{x})$ and the transition matrix $\boldsymbol{T}(\boldsymbol{x})$, where $\boldsymbol{T}_{ij}(\boldsymbol{x}) = P(\tilde{Y} = i|Y = j, X)$, *i.e.*, $P(\boldsymbol{Y}|X = \boldsymbol{x}) = [\boldsymbol{T}(\boldsymbol{x})]^{-1}P(\tilde{\boldsymbol{Y}}|X = \boldsymbol{x})$. Then, the expected risk of a function $f(X, Y)$ modeling $P(\boldsymbol{Y}|X)$ can be formulated as the expected risk of a function $g(X, \tilde{Y})$ modeling $P(\tilde{\boldsymbol{Y}}|X)$ multiplied by $\boldsymbol{T}(X)$, *i.e.*, $R(f(X, Y)) = R([\boldsymbol{T}(X)]^{-1}g(X, \tilde{Y}))$. In practice, the expected risk $R([\boldsymbol{T}(X)]^{-1}g(X, \tilde{Y}))$ can not be calculated, existing methods approximate the expected risk with the averaged loss over the noisy training examples. Normally, when the number of examples is limited, the variance of the losses or the empirical risk could be high, which could make the algorithm unstable and lead to a large estimation error.

Different variance reduction methods have been developed in many fields. For example, Truncated Importance Sampling (Ionides, 2008) limits the maximum importance weight, which solves the problem of infinite variance and decreases the mean squared estimation error of the standard importance sampling. Anschel et al. (2017) proposed to stabilize training procedure and improve performance by reducing approximation error variance in target rewards. Achab et al. (2015) proposed to use stochastic gradient descent (SGD) with variance reduction for optimizing a finite average of smooth convex functions, and a linear rate of convergence under strong convexity is proved. Similarly, Allen-Zhu & Hazan (2016) proved that a fast convergence rate can be achieved by using variance reduction on the non-convex optimization problem. Although the definitions of variance are different, those works motivate us to explore the role of variance of losses in learning with noisy labels because it is natural to consider that penalizing the variance of losses will have some benefits similar to previous works.

## 3    ENHANCING VARIANCE OF LOSSES FOR LEARNING WITH NOISY LABEL

In this section, we propose our method, *i.e.*, losses Variance Regularization for label-Noise Learning (VRNL). We reveal how the proposed method reduces the negative effects of incorrect labels. We also illustrate the advantage of our regularizer when working with existing methods.

### 3.1    METHODOLOGY

We show that the proposed method can efficiently prevent the model from learning incorrect labels. Intuitively, encouraging the variance of losses can prevent losses of instances with incorrect labels from decreasing, and promote the reduction of losses of instances with correct labels. Theoretically, the gradients of large-loss examples will be assigned with small weights, and gradients of small-loss examples will be assigned with large weights. As a result, the model puts more trust on small-loss examples; and small-loss examples will contribute more to the update of parameters, which could reduce the harmfulness of instances with incorrect labels.

To analyze the regularization effect in our method, we have to define some notations here. Let $\mathrm{Var}[.]$ denote variance of a distribution. For a random variable $X$, $\mathrm{Var}[X] = \mathbb{E}[X^2] - \mathbb{E}^2[X]$. Let $C$ denote the number of classes. Let $f_\theta : \mathcal{X} \to \Delta^{C-1}$ be a mapping parameterized by $\theta$ (*e.g.*, a neural network), where $\Delta^{C-1}$ denotes a probability simplex. Generally, the expected risk w.r.t. noisy data is formulated as $\mathbb{E}_{(X, \tilde{Y})}[\ell(f_\theta(X), \tilde{Y})]$, where $\ell(\cdot)$ is the loss function employed. Following analysis can be applied to all methods which can be formulated as $\mathbb{E}_{(X, \tilde{Y})}[\ell(f_\theta(X), \tilde{Y})]$, where $\ell(\cdot)$, including statistically-consistent methods and cross-entropy loss-based methods. We propose to add a variance regularizer to the losses. Specifically, the objective function of our method is

$$R_G(f_\theta) = \mathbb{E}_{(X, \tilde{Y})}[\ell(f_\theta(X), \tilde{Y})] - \alpha \mathrm{Var}_{(X, \tilde{Y})}[\ell(f_\theta(X), \tilde{Y})], \tag{1}$$

where $\mathrm{Var}_{(X, \tilde{Y})}[\ell(f_\theta(X), \tilde{Y})]$ is a regularization term, and $\alpha$ is an adjustable hyper-parameter to control the strength of the regularization effect. To encourage the variance of losses, $\alpha$ is chosen to be a positive value. Usually, the strength of the regularization effect should not be too large, and $\alpha$ is much smaller than 1. A suitable $\alpha$ can be obtained by utilizing validation sets. More details and discussions can be found in Appendix.

To exploit the influence of our designed regularizer with respect to the update of parameter $\theta$, we first illustrate the derivative of the objective function to $\theta$, *i.e.*,

$$\frac{R_G(f_\theta)}{\partial \theta} = \frac{\partial \mathbb{E}_{(X,\tilde{Y})}[\ell(f_\theta(X),\tilde{Y})]}{\partial \theta} - \alpha \frac{\partial \text{Var}_{(X,\tilde{Y})}[\ell(f_\theta(X),\tilde{Y})]}{\partial \theta} = \mathbb{E}_{(X,\tilde{Y})}\left[W\frac{\partial \ell(f_\theta(X),\tilde{Y})}{\partial \theta}\right], \quad (2)$$

where $W = 1 + 2\alpha\left(\mathbb{E}_{(X,\tilde{Y})}[\ell(f_\theta(X),\tilde{Y})] - \ell(f_\theta(X),\tilde{Y})\right)$. For a specific example $(x,\tilde{y})$, its corresponding gradient is $w\frac{\partial \ell(f_\theta(x),\tilde{y})}{\partial \theta}$, where the weight $w$ is $w = 1 + 2\alpha(\mathbb{E}_{(X,\tilde{Y})}[\ell(f_\theta(X),\tilde{Y})] - \ell(f_\theta(x),\tilde{y}))$. As aforementioned, $\alpha$ is chosen to be small such that $w$ should be positive. The above equation shows that 1) if the loss of the example $(x,\tilde{y})$ is smaller than the expectation of the losses, $\left(\mathbb{E}_{(X,\tilde{Y})}[\ell(f_\theta(X),\tilde{Y})] - \ell(f_\theta(x),\tilde{y})\right)$ will be positive, and the weight associated with its gradient is larger than 1. Then the example contributes more to the update of the parameter $\theta$. 2) If the loss of the example $(x,\tilde{y})$ is larger than the expectation of the losses, $\left(\mathbb{E}_{(X,\tilde{Y})}[\ell(f_\theta(X),\tilde{Y})] - \ell(f_\theta(x),\tilde{y})\right)$ will be negative. The weight associated with its gradient is small. Then the example contributes less to the update of parameter $\theta$.

Due to the memorization effect, deep neural networks tend to learn easy examples first and gradually learn hard examples (Han et al., 2018b; Arpit et al., 2017). In learning with noisy labels, large-loss examples are more likely to have incorrect labels and should not be trusted (Bai et al., 2021). By employing the proposed method, the gradients of examples with incorrect labels are assigned with small weights. In such a way, incorrectly-labeled examples would have less contribution to update the parameter $\theta$, which prevents the model from overfitting incorrect labels.

Additionally, compared with existing small-loss based methods, our method can sufficiently exploit the information contained in the whole training dataset. Existing learning with noisy labels methods (Han et al., 2018b; Nguyen et al., 2019) usually divide the training sample into confident examples and unconfident examples based on the small-loss trick (Jiang et al., 2018; Malach & Shalev-Shwartz, 2017; Li et al., 2020). To be specific, the examples with large losses are unconfident examples, and their labels are ignored. However, some of the unconfident examples are hard-clean examples that contain useful information for learning noise-robust classifiers (Bai & Liu, 2021). In contrast, our method does not ignore unconfident examples but assign them with small weights such that all the label information of the training dataset has been carefully exploited.

## 3.2 PRACTICAL IMPLEMENTATION

In practice, the expected risk $R_G(f_\theta)$ in Eq. 1 can not be calculated, the empirical risk is employed as an approximation. Let $n$ be the number of training examples, generally, the empirical risk of our method is as follows:

$$\hat{R}(f_\theta) = \frac{1}{n}\sum_{i=1}^{n}\ell(f_\theta(x_i),\tilde{y}_i) - \alpha\left(\frac{1}{n}\sum_{i=1}^{n}\ell(f_\theta(x_i),\tilde{y}_i)^2 - \left(\frac{1}{N}\sum_{i=1}^{n}\ell(f_\theta(x_i),\tilde{y}_i)\right)^2\right). \quad (3)$$

We further illustrate specific forms and settings of our designed regularization working with existing methods, *i.e.*, Importance Reweighting (Liu & Tao, 2015), Forward (Patrini et al., 2017), and VolMinNet (Li et al., 2021). Empirically, our method improves their classification accuracy.

**Work with Forward.** Forward correction (Patrini et al., 2017) exploits the noise transition matrix $\boldsymbol{T}$ to estimate the clean class posterior distribution. We use the same method in the original paper (Patrini et al., 2017) to estimate the transition matrix.

The objective loss function by combining our method with Forward can be formulated as follows:

$$\hat{R}_{\text{Forward}}(\theta,\hat{\boldsymbol{T}}) = \frac{1}{n}\sum_{i=1}^{n}\ell_{CE}(\hat{\boldsymbol{T}}f_\theta(x_i),\tilde{y}_i)$$
$$- \alpha\left(\frac{1}{n}\sum_{i=1}^{n}\ell_{CE}(\hat{\boldsymbol{T}}f_\theta(x_i),\tilde{y}_i)^2 - \left(\frac{1}{n}\sum_{i=1}^{n}\ell_{CE}(\hat{\boldsymbol{T}}f_\theta(x_i),\tilde{y}_i)\right)^2\right),$$

where $\ell_{CE}$ is the cross-entropy loss, $\hat{\boldsymbol{T}}$ is the estimated transition matrix, $f_\theta$ models the clean class-posterior distribution, $\hat{\boldsymbol{T}}f_\theta$ models the noisy class-posterior distribution.

**Work with Importance Reweighting.** Importance Reweighting uses the weighted empirical risk to estimate the empirical risk with respect to clean class-posterior distribution (Liu & Tao, 2015). To calculate the weight of the empirical risk, both noisy class-posterior distribution and clean class-posterior distribution need to be estimated. The objective loss function by combining our method with Important Reweighting can be formulated as follows:

$$\hat{R}_{IR}(f_\theta) = \frac{1}{n}\sum_{i=1}^{n} \hat{\beta}_i \ell_{CE}(f_\theta(x_i), \tilde{y}_i) - \alpha\left(\frac{1}{n}\sum_{i=1}^{n}\hat{\beta}_i^2 \ell_{CE}(f_\theta(x_i),\tilde{y}_i)^2 - \left(\frac{1}{n}\sum_{i=1}^{n}\hat{\beta}_i\ell_{CE}(f_\theta(x_i),\tilde{y}_i)\right)^2\right),$$

where $\hat{\beta}_i = \frac{\hat{P}_D(y_i|x_i)}{\hat{P}_{D_\rho}(\tilde{y}_i|x_i)}$, $D$ is the clean distribution, $D_\rho$ is the noisy distribution. The gradient of $\hat{R}_{IR}$ w.r.t. an example $(x_i, \tilde{y})$ is as follows:

$$\nabla \hat{R}_{IR}(f_\theta, (x,\tilde{y})) = \frac{1}{n}\sum_{i=1}^{n}\hat{w}_i\left(\ell_{CE}(f_\theta(x_i),\tilde{y}_i)\frac{\partial\hat{\beta}_i}{\partial\theta} + \hat{\beta}_i\frac{\partial\ell_{CE}(f_\theta(x_i),\tilde{y}_i)}{\partial\theta}\right),$$

where $\hat{w}_i = 1 + 2\alpha\left(\frac{1}{n}\sum_{j=1}^{n}\hat{\beta}_j\ell_{CE}(f_\theta(x_j),\tilde{y}_j) - \hat{\beta}_i\ell_{CE}(f_\theta(x_i),\tilde{y}_i)\right)$. The $\ell_{CE}(f_\theta(x),\tilde{y})\frac{\partial\hat{\beta}_i}{\partial\theta} + \hat{\beta}_i\frac{\partial\ell_{CE}(f_\theta(x),\tilde{y})}{\partial\theta}$ is the gradient of the original Importance Reweighting loss. When the label $\tilde{y}_i$ is incorrect, the reweighted loss $\hat{\beta}_i\ell_{CE}(f_\theta(x_i),\tilde{y}_i)$ is usually larger than the averaged loss $\frac{1}{n}\sum_{j=1}^{n}\hat{\beta}_j\ell_{CE}(f_\theta(x_j),\tilde{y}_j)$. Then their difference is negative, which lead the weight $\hat{w}_i$ to be small because the hyper-parameter $\alpha$ is positive. As a result, the instance with an incorrect label has a small contribution to the update of parameter $\theta$, the proposed method can prevent the model from memorizing the incorrect labels.

In the implementation, the early stopping technique is used for the approximation of the clean class-posterior distribution. Specifically, the model $f_\theta$ is trained on noisy data with 20 epochs, and we feed the model output to a softmax function, then use the output of the softmax function $g(x)$ to approximate the clean class-posterior distribution. The noise transition matrix $\boldsymbol{T}$ has also been estimated by using the same approach as in Forward correction. Then the model $f_\theta$ is further optimized by both weighted empirical risk and regularization for the variance of losses as follows:

$$\hat{R}_{IR}(\theta) = \frac{1}{n}\sum_{i=1}^{n}\left[\ell_{CE}(f_\theta(x_i),\tilde{y}_i)\frac{g_{\tilde{y}}(x_i)}{(\hat{\boldsymbol{T}}g)_{\tilde{y}}(x_i)}\right] - \alpha\hat{\sigma}_\theta^2,$$

where

$$\hat{\sigma}_\theta^2 = \left(\frac{1}{n}\sum_{i=1}^{n}\left(\ell_{CE}(f_\theta(x_i),\tilde{y}_i)\frac{g_{\tilde{y}}(x_i)}{(\hat{\boldsymbol{T}}g)_{\tilde{y}}(x_i)}\right)^2 - \left(\frac{1}{n}\sum_{i=1}^{n}\ell_{CE}(f_\theta(x_i),\tilde{y}_i)\frac{g_{\tilde{y}}(x_i)}{(\hat{\boldsymbol{T}}g)_{\tilde{y}}(x_i)}\right)^2\right).$$

**Work with VolMinNet.** VolMinNet is an end-to-end label-noise learning method that learns the transition matrix and the clean class-posterior distribution simultaneously (Li et al., 2021). It optimizes two objectives: 1) a trainable diagonally dominant column stochastic matrix $\hat{\boldsymbol{T}}$ by minimizing the determinate $\log\det(\hat{\boldsymbol{T}})$; 2) the parameter $\theta$ of the model by the cross-entropy loss between the noisy label and the predicted probability by the neural network. In experiments, our VRNL only regularizes the parameter $\theta$ by calculating the variance of cross-entropy losses. The objective by combining our method with VolMinNet can be formulated as follows:

$$\hat{R}_{vol}(\theta, \hat{\boldsymbol{T}}) = \frac{1}{n}\sum_{i=1}^{n}\ell_{CE}(\hat{\boldsymbol{T}}f_\theta(x_i), \tilde{y}_i) + \lambda\log\det(\hat{\boldsymbol{T}})$$

$$- \alpha\left(\frac{1}{n}\sum_{i=1}^{n}\ell_{CE}(\hat{\boldsymbol{T}}f_\theta(x_i),\tilde{y}_i)^2 - \left(\frac{1}{n}\sum_{i=1}^{n}\ell_{CE}(\hat{\boldsymbol{T}}f_\theta(x_i),\tilde{y}_i)\right)^2\right),$$

where $\lambda > 0$ is an adjustable hyper-parameter, we set $\lambda = 0.0001$ in all experiments. The transition matrix $\hat{\boldsymbol{T}}$ should be differentiable, diagonally dominant and column stochastic.

Our method could help the *state-of-the-art* method VolMinNet (Li et al., 2021) to better estimate the transition matrix and the clean class-posterior distribution. Specifically, VolMinNet requires the clean class-posteriors to be diverse, which is called the *sufficiently scattered* assumption (Li et al., 2021). By increasing the variance of the loss, the diversity of the estimated noisy class posteriors is encouraged, so the estimated clean class-posteriors are also encouraged. Then transition matrix can be better learned, which leads to the clean class-posterior distribution being better estimated.

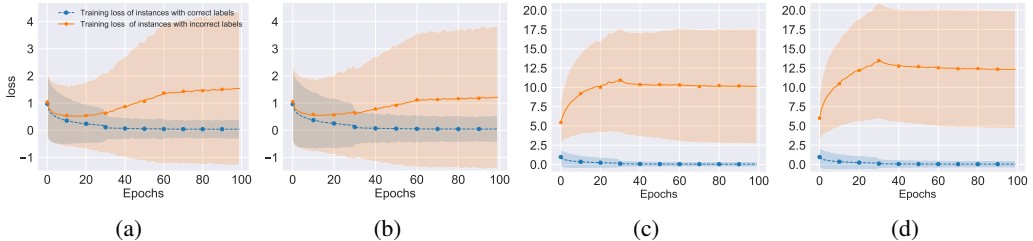

Figure 2: The change of losses with the increasing of training epochs for Reweighting. (a) and (b) illustrate CE losses of $P(Y|X)$ without or with increasing variance of losses, respectively. (c) and (d) illustrate CE losses of $P(\tilde{Y}|X)$ without or with increasing variance of losses, respectively.

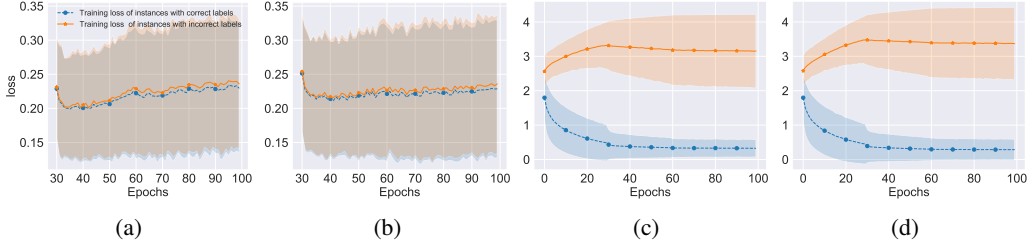

Figure 3: The change of losses with the increasing of training epochs for VolMinNet. (a) and (b) illustrate CE losses of $P(Y|X)$ without or with increasing variance of losses, respectively. (c) and (d) illustrate CE losses of $P(\tilde{Y}|X)$ without or with increasing variance of losses, respectively.

## 4 EXPERIMENTS

In this section, we first illustrate the empirical results of VRNL and other baselines on both synthetic and real-world noisy datasets. What is more, we will delve into the different effects of the proposed method on correct and incorrect examples to verify its effectiveness on Sec. 3. Finally, we will illustrate the robustness of the proposed method when the estimated transition matrix is biased.

**Datasets.** We verify the performance of proposed method on the manually corrupted version of three datasets, *i.e.*, MNIST (LeCun et al., 2010), CIFAR-10 (Krizhevsky et al., 2009) and CIFAR-100 (Krizhevsky et al., 2009), and one real-world noisy dataset, *i.e.*, Clothing1M (Xiao et al., 2015). We leave out 10% of training data as validation sets. The experiments are repeated five times on the synthetic noisy datasets. Clothing1M (Xiao et al., 2015) contains 1M images with real-world noisy labels, it also contains 50k, 14k, and 10k images with clean labels for training, validation, and testing, respectively. Existing methods like Forward (Patrini et al., 2017) and T-revision (Xia et al., 2019) use the 50k clean data to initialize the transition matrix and validate on 14k clean data. We assume that the clean data is not accessible, therefore, the clean data are not used for training and validation. We leave out 10% of examples from 1M noisy data for validation.

**Baselines.** The baselines used in our experiments: 1). CE, standard cross-entropy loss; 2). Decoupling (Malach & Shalev-Shwartz, 2017) trains two models at the same time, and only the instances which have different predictions from two networks are used to update the parameter; 3). MentorNet (Jiang et al., 2018) pre-trains an extra model which is used to select clean examples for the main model training; 4). Co-teaching (Han et al., 2018b) trains two networks simultaneously, and each network is used to select small-loss examples as trust examples to its peer network for further training; 5). Forward (Patrini et al., 2017) estimates the transition matrix in advance, then uses it to approximate the clean class posteriors; 6). T-Revision (Xia et al., 2019) proposes a method to fine-tune the estimated transition matrix to improve the classification performance; 7). Dual T (Yao et al., 2020b) improves the estimation of the transition matrix by introducing an intermediate class, and then factorizes the transition matrix into the product of two easy-to-estimate transition matrices; 8). VolMinNet (Li et al., 2021) is an end-to-end label-noise learning method, which can learn the transition matrix and the classifier simultaneously; 9). Reweight (Liu & Tao, 2015) uses the importance reweighting technique to estimate the expected risk on the clean domain by using noisy data. Note that the aim of this paper is not to design a *state-of-the-art* noisy-label learning algorithm, we want to explore whether the variance of losses should be always penalized when learning with noisy labels. Therefore, we do not make comparisons with some methods which use the semi-supervised methods, such as DevideMix (Li et al., 2020) and PES (Bai et al., 2021).

Table 1: Means and standard deviations (percentage) of classification accuracy. Results with "*" mean that they are the highest accuracy.

| | MNIST | | CIFAR-10 | | CIFAR-100 | |
|---|---|---|---|---|---|---|
| | Sym-20% | Sym-50% | Sym-20% | Sym-50% | Sym-20% | Sym-50% |
| Decoupling | $97.04 \pm 0.06$ | $94.58 \pm 0.08$ | $77.32 \pm 0.35$ | $54.07 \pm 0.46$ | $41.92 \pm 0.49$ | $22.63 \pm 0.44$ |
| MentorNet | $97.21 \pm 0.06$ | $95.56 \pm 0.15$ | $81.35 \pm 0.23$ | $73.47 \pm 0.15$ | $42.88 \pm 0.41$ | $32.66 \pm 0.40$ |
| Co-teaching | $97.07 \pm 0.10$ | $95.20 \pm 0.23$ | $82.27 \pm 0.07$ | $75.55 \pm 0.07$ | $48.48 \pm 0.66$ | $36.77 \pm 0.52$ |
| T-Revision | $98.72 \pm 0.10$ | $98.23 \pm 0.10$ | $87.95 \pm 0.36$ | $80.01 \pm 0.62$ | $62.72 \pm 0.69$ | $49.12 \pm 0.22$ |
| Dual T | $98.43 \pm 0.05$ | $98.15 \pm 0.12$ | $88.35 \pm 0.33$ | $82.54 \pm 0.19$ | $62.16 \pm 0.58$ | $52.49 \pm 0.37$ |
| CE | $98.65 \pm 0.05$ | $97.94 \pm 0.17$ | $86.86 \pm 0.26$ | $78.93 \pm 0.47$ | $60.15 \pm 0.46$ | $45.66 \pm 0.68$ |
| CE-VRNL | $\mathbf{98.71 \pm 0.06}$ | $\mathbf{97.96 \pm 0.06}$ | $\mathbf{87.49 \pm 0.25}$ | $\mathbf{79.31 \pm 0.52}$ | $\mathbf{60.38 \pm 0.61}$ | $\mathbf{47.79 \pm 0.52}$ |
| Forward | $97.47 \pm 0.15$ | $97.93 \pm 0.22$ | $87.29 \pm 0.63$ | $77.58 \pm 1.05$ | $59.71 \pm 0.40$ | $44.53 \pm 1.11$ |
| Forward-VRNL | $\mathbf{98.89 \pm 0.04}^*$ | $\mathbf{98.14 \pm 0.27}$ | $\mathbf{89.81 \pm 0.29}^*$ | $\mathbf{81.16 \pm 0.55}$ | $\mathbf{68.19 \pm 0.31}^*$ | $\mathbf{54.10 \pm 1.2}$ |
| Reweight | $98.20 \pm 0.24$ | $97.93 \pm 0.20$ | $88.42 \pm 0.18$ | $82.13 \pm 0.56$ | $60.52 \pm 0.52$ | $47.69 \pm 0.78$ |
| Reweight-VRNL | $\mathbf{98.61 \pm 0.21}$ | $\mathbf{98.27 \pm 0.15}^*$ | $\mathbf{89.68 \pm 0.24}$ | $\mathbf{83.99 \pm 0.28}^*$ | $\mathbf{66.52 \pm 0.25}$ | $\mathbf{50.26 \pm 0.14}$ |
| VolMinNet | $98.66 \pm 0.14$ | $97.83 \pm 0.15$ | $89.27 \pm 0.30$ | $82.17 \pm 0.19$ | $65.65 \pm 0.62$ | $54.40 \pm 0.62$ |
| VolMinNet-VRNL | $\mathbf{98.78 \pm 0.08}$ | $\mathbf{97.93 \pm 0.20}$ | $\mathbf{89.42 \pm 0.12}$ | $\mathbf{82.92 \pm 0.24}$ | $\mathbf{66.40 \pm 0.66}$ | $\mathbf{55.94 \pm 0.64}^*$ |
| | Asym-20% | Asym-50% | Asym-20% | Asym-50% | Asym-20% | Asym-50% |
| Decoupling | $96.79 \pm 0.01$ | $94.71 \pm 0.08$ | $78.63 \pm 0.27$ | $71.01 \pm 3.72$ | $39.42 \pm 0.48$ | $21.64 \pm 0.23$ |
| MentorNet | $97.03 \pm 0.05$ | $94.66 \pm 0.11$ | $78.99 \pm 0.34$ | $68.00 \pm 2.09$ | $10.03 \pm 0.33$ | $11.14 \pm 0.25$ |
| Co-teaching | $97.02 \pm 0.03$ | $95.15 \pm 0.09$ | $83.96 \pm 0.28$ | $76.58 \pm 0.84$ | $13.36 \pm 0.44$ | $13.10 \pm 0.66$ |
| T-Revision | $98.90 \pm 0.11$ | $\mathbf{98.35 \pm 0.13}^*$ | $88.38 \pm 0.56$ | $81.51 \pm 0.74$ | $59.52 \pm 0.43$ | $45.56 \pm 1.86$ |
| Dual T | $95.46 \pm 0.14$ | $91.46 \pm 0.29$ | $70.31 \pm 1.06$ | $53.04 \pm 2.76$ | $05.80 \pm 0.78$ | $02.38 \pm 0.94$ |
| CE | $98.76 \pm 0.07$ | $97.91 \pm 0.23$ | $87.31 \pm 0.32$ | $79.47 \pm 0.47$ | $59.83 \pm 0.69$ | $45.08 \pm 0.71$ |
| CE-VRNL | $\mathbf{98.77 \pm 0.12}$ | $\mathbf{97.98 \pm 0.20}$ | $\mathbf{87.38 \pm 0.31}$ | $\mathbf{79.61 \pm 0.40}$ | $\mathbf{60.69 \pm 0.51}$ | $\mathbf{45.44 \pm 0.10}$ |
| Forward | $98.42 \pm 0.06$ | $97.92 \pm 0.04$ | $87.70 \pm 0.29$ | $79.25 \pm 1.61$ | $60.24 \pm 0.42$ | $43.39 \pm 1.15$ |
| Forward-VRNL | $\mathbf{98.92 \pm 0.06}^*$ | $\mathbf{98.13 \pm 0.21}$ | $\mathbf{89.98 \pm 0.11}^*$ | $\mathbf{82.35 \pm 0.88}$ | $\mathbf{67.89 \pm 0.30}^*$ | $\mathbf{53.67 \pm 0.52}$ |
| Reweight | $98.50 \pm 0.07$ | $98.09 \pm 0.08$ | $88.55 \pm 0.32$ | $82.72 \pm 0.38$ | $60.81 \pm 0.70$ | $46.36 \pm 0.18$ |
| Reweight-VRNL | $\mathbf{98.77 \pm 0.21}$ | $\mathbf{98.10 \pm 0.16}$ | $\mathbf{89.80 \pm 0.11}$ | $\mathbf{84.20 \pm 0.25}^*$ | $\mathbf{66.62 \pm 0.45}$ | $\mathbf{49.71 \pm 0.64}$ |
| VolMinNet | $98.62 \pm 0.10$ | $98.03 \pm 0.12$ | $89.50 \pm 0.18$ | $83.15 \pm 0.56$ | $66.02 \pm 0.73$ | $55.17 \pm 0.46$ |
| VolMinNet-VRNL | $\mathbf{98.76 \pm 0.13}$ | $\mathbf{98.08 \pm 0.08}$ | $\mathbf{89.64 \pm 0.19}$ | $\mathbf{83.65 \pm 0.32}$ | $\mathbf{66.24 \pm 0.95}$ | $\mathbf{55.85 \pm 0.73}^*$ |
| | Pair-20% | Pair-45% | Pair-20% | Pair-45% | Pair-20% | Pair-45% |
| Decoupling | $96.93 \pm 0.07$ | $94.34 \pm 0.54$ | $77.12 \pm 0.30$ | $53.71 \pm 0.99$ | $40.12 \pm 0.26$ | $27.97 \pm 0.12$ |
| MentorNet | $96.89 \pm 0.04$ | $91.98 \pm 0.46$ | $77.42 \pm 0.23$ | $61.03 \pm 0.20$ | $39.22 \pm 0.47$ | $26.48 \pm 0.37$ |
| Co-teaching | $97.00 \pm 0.06$ | $96.25 \pm 0.01$ | $80.65 \pm 0.20$ | $73.02 \pm 0.23$ | $42.79 \pm 0.79$ | $27.97 \pm 0.20$ |
| T-Revision | $98.89 \pm 0.08$ | $84.56 \pm 8.18$ | $90.33 \pm 0.52$ | $78.94 \pm 2.58$ | $64.33 \pm 0.49$ | $41.55 \pm 0.95$ |
| Dual T | $98.86 \pm 0.04$ | $96.71 \pm 0.12$ | $89.77 \pm 0.25$ | $76.53 \pm 2.51$ | $67.21 \pm 0.43$ | $47.60 \pm 0.43$ |
| CE | $98.71 \pm 0.08$ | $83.49 \pm 3.77$ | $88.63 \pm 0.26$ | $66.32 \pm 2.44$ | $\mathbf{61.04 \pm 0.31}$ | $39.78 \pm 0.30$ |
| CE-VRNL | $\mathbf{98.80 \pm 0.10}$ | $\mathbf{84.00 \pm 3.65}$ | $\mathbf{88.71 \pm 0.30}$ | $\mathbf{67.71 \pm 2.04}$ | $61.00 \pm 0.32$ | $\mathbf{39.91 \pm 0.20}$ |
| Forward | $98.85 \pm 0.09$ | $96.45 \pm 4.03$ | $\mathbf{90.88 \pm 0.23}^*$ | $83.27 \pm 9.47$ | $62.54 \pm 0.42$ | $41.96 \pm 1.45$ |
| Forward-VRNL | $\mathbf{98.88 \pm 0.08}$ | $\mathbf{96.55 \pm 3.88}$ | $\mathbf{90.88 \pm 0.29}^*$ | $\mathbf{83.54 \pm 9.29}$ | $\mathbf{62.78 \pm 0.32}$ | $\mathbf{42.29 \pm 1.23}$ |
| Reweight | $98.64 \pm 0.07$ | $95.52 \pm 3.58$ | $89.68 \pm 0.30$ | $76.03 \pm 5.02$ | $61.35 \pm 0.66$ | $40.10 \pm 0.46$ |
| Reweight-VRNL | $\mathbf{98.68 \pm 0.14}$ | $\mathbf{95.97 \pm 3.52}$ | $\mathbf{89.83 \pm 0.30}$ | $\mathbf{76.75 \pm 6.15}$ | $\mathbf{61.37 \pm 0.42}$ | $\mathbf{40.30 \pm 0.57}$ |
| VolMinNet | $\mathbf{99.05 \pm 0.05}^*$ | $99.08 \pm 0.06$ | $90.73 \pm 0.23$ | $88.47 \pm 0.61$ | $69.96 \pm 1.18$ | $61.85 \pm 1.41$ |
| VolMinNet-VRNL | $99.02 \pm 0.08$ | $\mathbf{99.10 \pm 0.08}^*$ | $\mathbf{90.86 \pm 0.27}$ | $\mathbf{88.77 \pm 0.51}^*$ | $\mathbf{70.18 \pm 0.50}^*$ | $\mathbf{63.38 \pm 1.72}^*$ |

Table 2: Classification accuracy(percentage) on Clothing1M. Only noisy data are exploited for training and validation.

| Decoupling | MentorNet | Co-teaching | T-Revision | Dual T | PTD |
|---|---|---|---|---|---|
| 54.53 | 56.79 | 60.15 | 70.97 | 70.17 | 71.67 |
| Forward | Forward-VRNL | Reweight | Reweight-VRNL | VolMinNet | VolMinNet-VRNL |
| 71.27 | **72.43** | 71.62 | **72.14** | 72.29 | **72.66** |

**Noise Types.** To generate a noisy dataset, we corrupted the training and validation sets manually according to a special transition matrix $\boldsymbol{T}$. Specifically, we conduct experiments on synthetic noisy datasets with three widely used types of noise: 1). Symmetry flipping (Sym-$\epsilon$) (Patrini et al., 2017); 2). Asymmetry flipping (Asym-$\epsilon$); 3). Pair flipping (Pair-$\epsilon$) (Han et al., 2018b). We manually corrupt the labels of instances according to the transition matrix $\boldsymbol{T}$.

**Network structure and optimization.** We implement the proposed methods and baseline using Pytorch 1.9.1 and train the models on TITAN Xp. The model structure and optimizer are as same as the *state-of-the-art* method (Li et al., 2021). Specifically, we use a LeNet-5 network (LeCun et al., 1998) for MNIST, a ResNet-18 network for CIFAR-10, a ResNet-32 network (He et al., 2016) for CIFAR-100, a ResNet-50 pretrained on ImageNet for Clothing 1M. On synthetic noise datasets,

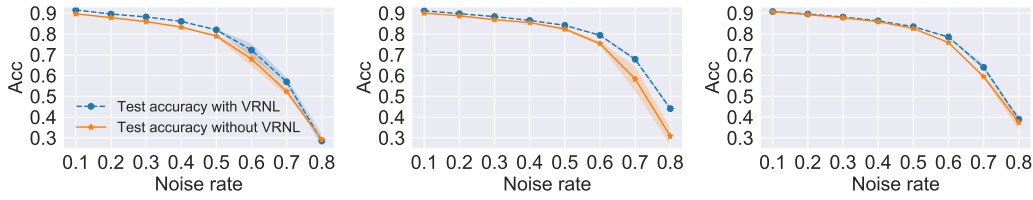

(a) Forward with VRNL  (b) Reweight with VRNL  (c) VolMinNet with VRNL

Figure 4: Test accuracies of the models trained on CIFAR10 with symmetry-flipping noise and increasing noise rate.

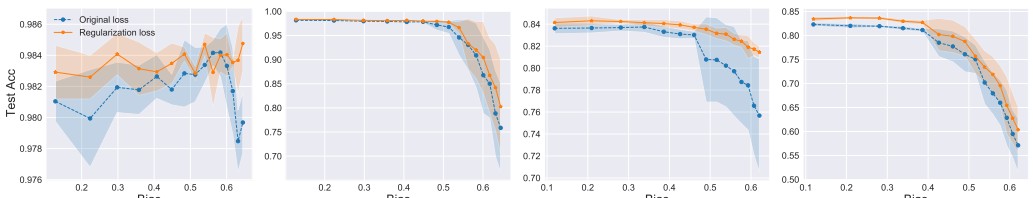

(a) Reweight on MNIST  (b) Forward on MNIST  (c) Reweight on CIFAR10  (d) Forward on CIFAR10

Figure 5: Test accuracies of the models trained on MNIST and CIFAR10 by using biased transition matrices, We increase the error of transition matrices manually. The proposed VRNL is robust to the biased transition matrix.

SGD is used to train the neural network with batch size 128, momentum 0.9, weight decay $10^{-4}$, and an initial learning rate $10^{-2}$. The algorithm is trained for 80 epochs, and the learning rate is divided by 10 after the 30-th and 60-th epochs. For Forward and Reweight, we set the hyper-parameter $\alpha = 0.1$ on symmetry-flipping noise, asymmetry-flipping noise, and we set $\alpha = 0.01$ on pair-flipping noise. For VolMinNet, we set $\alpha = 0.005$ on MNIST and CIFAR-100 with pair-flipping noise. For other experiments on synthetic noisy datasets, $\alpha = 0.05$ is employed. When the dataset is Clothing-1M, for Forward and Reweight, SGD with batch size 64, momentum 0.9, weight decay $10^{-4}$ is used to train the model, and $\alpha$ is set to be 0.1; for VolMinNet, SGD with batch size 64, momentum 0.9, weight decay $10^{-3}$ is used, and $\alpha$ is set to be 0.005. For Forward and Reweight, the transition matrix $T$ has to be estimated in advance. For the end-to-end method VolMinNet, the transition matrix $T$ and the classifier are learned simultaneously. To estimate the transition matrix, we follow the same experimental settings described in their original papers (Patrini et al., 2017; Li et al., 2021). The parameters of the model used to estimate the transition matrix will be used to initialize the weights of the classifier.

## 4.1 CLASSIFICATION ACCURACY EVALUATION

We embed VRNL into the label-noise learning methods, *e.g.*, Forward, Reweight and VolMin-Net which are named Forward-VRNL, Reweight-VRNL and VolMinNet-VRNL, respectively. In Tab. 1, we illustrate classification accuracies on datasets containing symmetry-flipping, asymmetry-flipping and pair-flipping noise. It shows that VRNL improves the classification accuracies of all the label-noise learning methods on different datasets and different types of noise. The performance of VolMinNet is usually better than that of Forward and Reweight and the estimation error of transition matrix used in Forward and Reweight is larger than in VolMinNet (Li et al., 2021). However, by employing VRNL, the performance of Forward and Reweight are comparable to that of VolMinNet, which suggests that VRNL is robust to the biased transition matrix. The improvements of VRNL under pair-flipping noise are not large compared with symmetry-flipping and asymmetry-flipping noise. More experiments to analyse the reason can be found in the Appendix.

We also provide the performance of VRNL under various noise rates. The experiment results are visualized in Fig. 4. VRNL can improve the performance of existing methods on both little noise and extreme noise. Detailed digits are posed in Tab. 4 and Tab. 5 in Appendix.

In Tab. 2, we illustrate the results on the real-world dataset Clothing1M. VRNL improves the generalization ability of backbone methods. The performance of VolMinNet-VRNL outperforms all other baselines.

### 4.2 THE INFLUENCE ON CLEAN AND NOISY CLASS POSTERIORS

To analyze the influence of variance of losses increase on clean class posteriors and noisy class posteriors. In Fig. 2 and Fig. 3, we visualize the change of cross-entropy losses for instances with clean labels and instances with noisy labels during the model training, respectively. The average loss and the standard derivation of mislabeled examples and correctly labeled examples are visualized separately for better illustration. The methods used are Reweight and VolMinNet.

By comparing Reweight-VRNL with Reweight, the loss of noisy labels for mislabeled examples is larger but the loss of noisy labels for correctly labeled examples is almost unchanged as shown in Fig. 2c and Fig. 2d. It means that the proposed method prevents the model from memorizing incorrect labels and has little influence on learning correctly labeled examples. By comparing Fig. 2b with Fig. 2a, the loss of clean labels for mislabeled examples becomes smaller when our method is employed. It implies that our method helps learn clean class posteriors of mislabeled examples.

Similarly, the results also hold for VolMinNet. Specifically, the variance of noisy class posteriors in Fig. 3d increases compared with Fig. 3c, which could help VolMinNet better estimate the transition matrix. It is because our method encourages the diversity of noisy class posteriors, which makes the *sufficiently scattered* assumption easier be satisfied when the sample size is limited. Meanwhile, it can be seen that the empirical risk defined by clean training examples decreases after using our method, as shown in Fig. 3a and Fig. 3b. It means that the model can classify the examples better.

### 4.3 PERFORMANCE WITH THE BIASED TRANSITION MATRIX

In practice, the noise transition matrix generally is not given and is required to be estimated. However, the estimated transition matrix could contain a large estimation error. One reason is that the transition matrix can be hard to accurately estimate when sample size is limited (Yao et al., 2020b). Another reason is that the assumptions (Patrini et al., 2017; Li et al., 2021) used to identify the transition matrix may not hold. This motivates us to investigate the performance of our regularizer when the transition matrix contains bias.

To simulate the estimation error, we manually inject noise into the transition matrix, *i.e.*, $\boldsymbol{T}^\rho = \boldsymbol{T} + \gamma|\Delta|$, where $\Delta \in \mathbb{R}^{C \times C}$ sampled from standard multivariate normal distribution, and $\gamma \in [0.01, 0.15]$. Then we normalize the column of the transition matrix $\boldsymbol{T}_\rho$ sum up to 1 by $\boldsymbol{T}_{ij}^N = \boldsymbol{T}^\rho{}_{ij} / \sum_{k=1}^{C} \boldsymbol{T}^\rho{}_{ik}$. The estimation error $\epsilon_T$ of a transition matrix is calculated by employing the entry-wise matrix norm, *i.e.*, $\epsilon_{\boldsymbol{T}} = \|\boldsymbol{T} - \boldsymbol{T}^N\|_{1,1} / \|\boldsymbol{T}\|_{1,1}$.

The biased transition matrix $T^N$ is adopted to Reweight, Reweight-VRNL, Forward and Forward-VRNL, respectively. Experimental results shown in Fig. 5 illustrate that our method is more robust to the bias transition matrix. Specifically, for most experiments and different levels of bias $\epsilon_T$, the test accuracies of Reweight-VRNL and Forward-VRNL are higher than Reweight and Forward. Additionally, the test accuracy of Reweight-VRNL drops much slower than Reweight with the increasing of bias $\epsilon_T$.

## 5 CONCLUSION

In this paper, we study whether we should always penalize the variance of losses for the problem of learning with noisy labels. Interestingly, we found that increasing the variance of losses could be helpful, which can boost the memorization effect and reduce the harmfulness of incorrect labels. Theoretically, we show that increasing variance of losses can reduce the weights of the gradient with respect to instances with incorrect labels, therefore these instances have a small contribution to the update of model parameters. A simple and effective method VRNL is also proposed which can be easily integrated into existing label-noise learning methods to improve their robustness. The experimental results on both synthetic and real-world noisy datasets demonstrate that VRNL can dramatically improve the performance of existing label-noise learning methods. Empirically, we have shown that the proposed method can help models better learn clean class posteriors. We have also illustrated that VRNL can improve the classification performance of existing methods when the transition matrix is poorly estimated, which makes our method be practically useful.

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

## A  THE GRADIENTS OF VRNL

The full derivation of the gradients of VRNL.

$$
\begin{aligned}
\frac{R_G(f_\theta)}{\partial \theta} &= \frac{\partial \mathbb{E}_{(X,\tilde{Y})}[\ell(f_\theta(X),\tilde{Y})]}{\partial \theta} - \alpha \frac{\partial Var_{(X,\tilde{Y})}[\ell(f_\theta(X),\tilde{Y})]}{\partial \theta} \\
&= \frac{\partial \mathbb{E}_{(X,\tilde{Y})}[\ell(f_\theta(X),\tilde{Y})]}{\partial \theta} - \alpha \left\{ \frac{\partial \mathbb{E}_{(X,\tilde{Y})}[\ell^2(f_\theta(X),\tilde{Y})]}{\partial \theta} - \frac{\partial \mathbb{E}_{(X,\tilde{Y})}^2[\ell(f_\theta(X),\tilde{Y})]}{\partial \theta} \right\} \\
&= \mathbb{E}_{(x,\tilde{y})\sim D_\rho}[\frac{\partial \ell(f_\theta(X),\tilde{Y})}{\partial \theta}] - \alpha \left\{ \mathbb{E}_{(x,\tilde{y})\sim D_\rho} \left[ 2\ell(f_\theta(X),\tilde{Y}) \frac{\partial \ell(f_\theta(X),\tilde{Y})}{\partial \theta} \right] \right. \\
&\quad \left. -2\mathbb{E}_{(x,\tilde{y})\sim D_\rho}[\ell(f_\theta(X),\tilde{Y})]\mathbb{E}_{(x,\tilde{y})\sim D_\rho} \left[ \frac{\partial \ell(f_\theta(X),\tilde{Y})}{\partial \theta} \right] \right\} \\
&= \mathbb{E}_{(x,\tilde{y})\sim D_\rho} \left[ \left( 1 + 2\alpha \mathbb{E}_{(x,\tilde{y})\sim D_\rho}[\ell(f_\theta(X),\tilde{Y})] - 2\alpha\ell(f_\theta(X),\tilde{Y}) \right) \frac{\partial \ell(f_\theta(X),\tilde{Y})}{\partial \theta} \right] \\
&= \mathbb{E}_{(X,\tilde{Y})} \left[ W \frac{\partial \ell(f_\theta(X),\tilde{Y})}{\partial \theta} \right],
\end{aligned}
\tag{4}
$$

where

$$
W = 1 + 2\alpha \left( \mathbb{E}_{(X,\tilde{Y})}[\ell(f_\theta(X),\tilde{Y})] - \ell(f_\theta(X),\tilde{Y}) \right).
\tag{5}
$$

# B    DISCUSSIONS ABOUT THE MEMORIZATION EFFECT

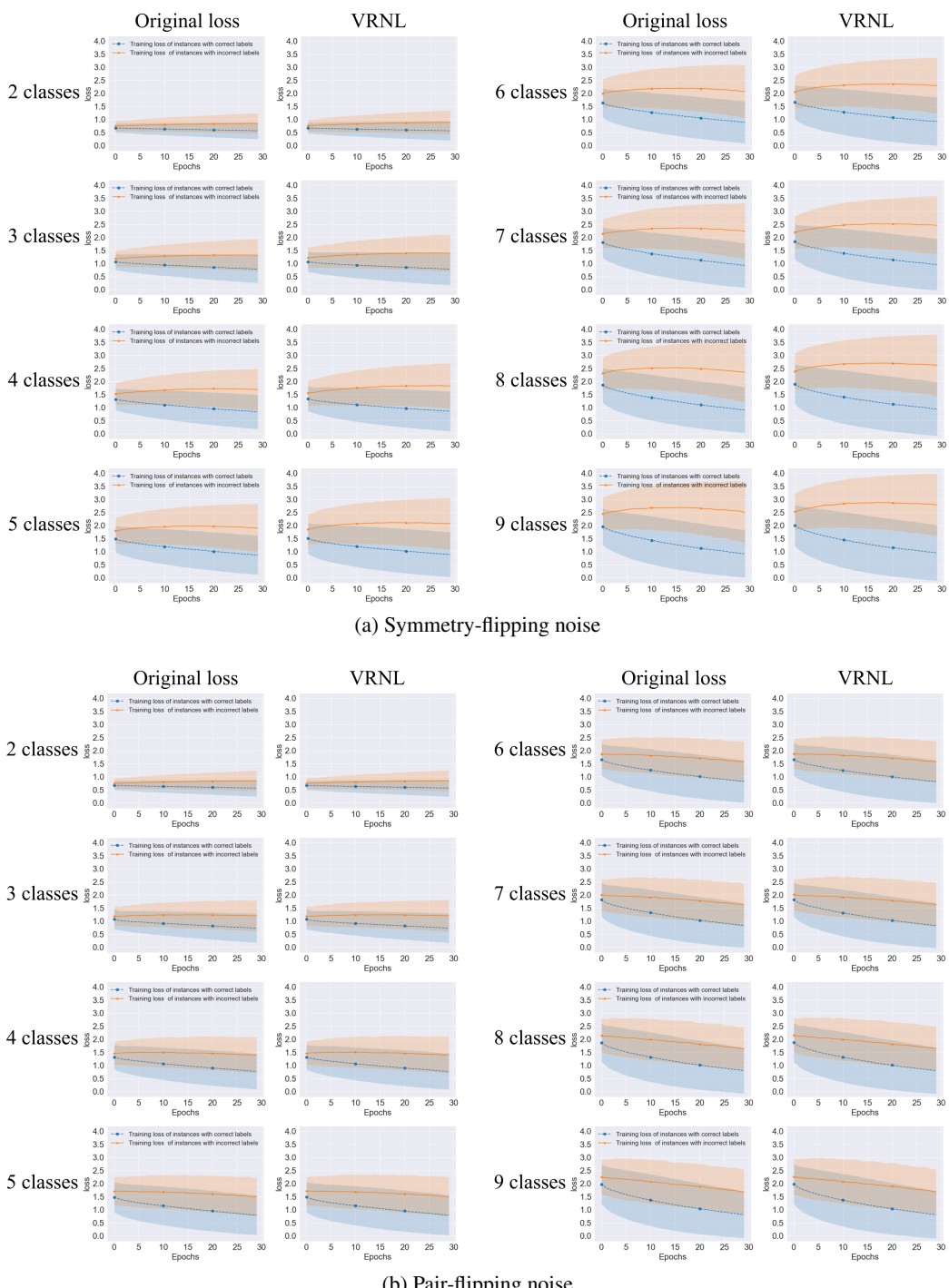

Figure 6: We increase the number of classes gradually, and the differences between the loss of instances with incorrect labels and the loss of instances with correct labels in symmetry-flipping noise are larger than the differences in pair-flipping noise. The differences are increasing gradually with the increase in the number of classes.

To investigate why the improvement of VRNL on pair-flipping noise is smaller than on symmetry-flipping noise, we conduct a series of experiments. We train a ResNet-18 (He et al., 2016) using

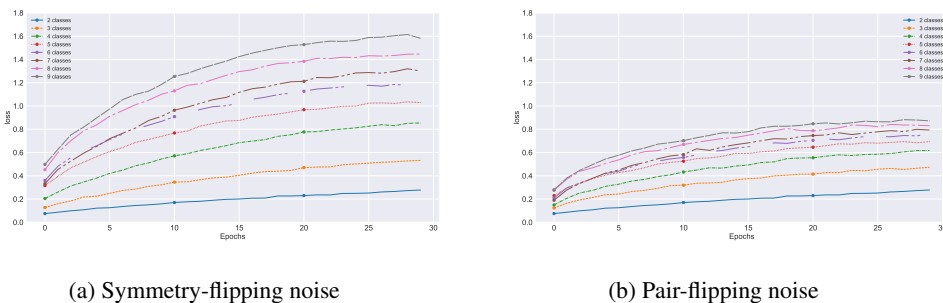

(a) Symmetry-flipping noise            (b) Pair-flipping noise

Figure 7: The differences between the loss of instances with incorrect labels and the loss of instances with correct labels in symmetry-flipping noise and pair-flipping noise.

Forward loss (Patrini et al., 2017) under the symmetry-flipping noise or pair flipping noise. The noise rate is 0.2. We reclassify 100 classes of CIFAR100 (Krizhevsky et al., 2009), *e.g.*, when the number of classes is 2, the classes of 0-49 are reclassified as 0, and the classes of 50-100 are reclassified as 1. The experiment results are shown in Fig. 6.

By comparing with the data containing pair flipping noise, we found that the memorization effect is stronger in the data containing symmetry-flipping noise. Then the improvement of VRNL on pair-flipping noise is smaller than symmetry-flipping noise because our method relies on the memorization effect. Specifically, the results show that the difference between the loss of instances with incorrect labels and the loss of instances with correct labels becomes larger in both symmetry-flipping noise and pair-flipping noise. However, by comparing with the average difference under the pair-flipping noise, the difference between the loss of instances with correct labels and the loss of instances with correct labels is larger under symmetry-flipping noise.

It implies that under symmetry-flipping noise, the memorization effect is strong because it is much easier to memorize the easy examples with correct labels than the hard examples with incorrect labels. We think the reason that the memorization effect is strong on symmetry-flipping noise is that in a noisy class, the contained noisy examples are from different classes, then memorizing all these noisy examples can be hard. By contrast, for pair-flipping noise, in a noisy class, the contained noisy examples are from one class, then it is easy for the learning model to find the common features and memorize these examples.

## C  EXPERIMENTS ON WEBVISION

We also conduct experiments on WebVision dataset 1.0 (Li et al., 2017). Following previous work (Chen et al., 2019), we train models on Google image subset and test model on Validation set. We first resize images to make shorter size as 320, then randomly crop a patch of image whose size is 299x299. Horizontal random flipping is used. The network structure is inception-resnet v2 (Szegedy et al., 2017). To estimate transition matrix for Forward and Reweight, we train the network for 20 epochs. We use SGD with momentum as 0.9, learning rate as 0.01, weight decay as $10^{-3}$. Then we follow the previous work (Liu & Tao, 2015; Patrini et al., 2017) using anchor point assumption to estimate transition matrix. We train the classifier for 80 epochs, learning rate is divided by 10 after 30 and 60 epochs. We start increasing variance of losses at 30th epoch. $\alpha$ is set to 0.05 for Forward-VRNL and VolMinNet-VRNL. For Reweight-VRNL, $\alpha$ is set to 0.005. The experiment results are provided at Tab. 3.

Table 3: The experiment results on WebVision.

|              | Forward | Reweight | VolMinNet |
| ------------ | ------- | -------- | --------- |
| Without VRNL | 33.40   | 46.28    | 72.44     |
| With VRNL    | **35.00** | **48.44** | **72.92** |

## D    EXPERIMENTS UNDER EXTREME NOISE

We also conduct the experiments under extreme noise, the noise rate is 80%, noise type is symmetry-flipping.

Table 4: The test accuracy under Sym-80% noise.

|  | Forward | Forward-VRNL | Reweight | Reweight-VRNL | VolMinNet | VolMinNet-VRNL |
|---|---|---|---|---|---|---|
| MNIST | $91.66 \pm 0.68$ | $\mathbf{92.46 \pm 2.13}$ | $93.09 \pm 1.79$ | $\mathbf{94.49 \pm 0.52}$ | $92.16 \pm 0.90$ | $\mathbf{92.35 \pm 0.25}$ |
| CIFAR10 | $28.16 \pm 1.86$ | $\mathbf{29.27 \pm 2.13}$ | $27.64 \pm 3.43$ | $\mathbf{43.56 \pm 4.13}$ | $37.10 \pm 3.33$ | $\mathbf{37.73 \pm 3.87}$ |
| CIFAR100 | $16.84 \pm 1.31$ | $\mathbf{19.87 \pm 0.67}$ | $6.64 \pm 1.84$ | $\mathbf{12.26 \pm 0.89}$ | $22.56 \pm 0.39$ | $\mathbf{23.84 \pm 1.17}$ |

The experiment results show that the proposed method still can improve the performance of Forward, Reweight and VolMinNet when the noise rate is large.

## E    EXPERIMENTS UNDER LITTLE NOISE

We also conduct the experiments under little noise on CIFAR-10, the noise rate is 10%, noise type is symmetry-flipping. The experiments results are shown in Table 5.

Table 5: The test accuracy under Sym-10% noise.

|  | Forward | Reweight | VolMinNet |
|---|---|---|---|
| Without VRNL | $89.74 \pm 0.27$ | $90.08 \pm 0.51$ | $90.49 \pm 0.09$ |
| With VRNL | $\mathbf{91.30 \pm 0.15}$ | $\mathbf{91.11 \pm 0.19}$ | $\mathbf{90.77 \pm 0.13}$ |

The experiment results show that the proposed method still can improve the performance of Forward, Reweight and VolMinNet when the noise rate is small.

## F    EXPERIMENTS ON CLEAN DATASETS

We also conduct the experiment on clean datasets using standard cross entropy loss with VRNL, $\alpha$ is set to 0.01. The experiment results are shown in Tab. 6.

The VRNL has little negative influence when the dataset is clean.

## G    SENSITIVITY ANALYSIS

We conduct the sensitivity analysis on one synthetic dataset, CIFAR-10, under symmetry-flipping noise, the noise rate is 50%. The $\alpha$ increases from 0.001 to 0.3. The experiment results are shown in Fig. 8. Overall, the curve is smooth, thus VRNL is not sensitive. We also show corresponding validation accuracy. As shown in Fig. 9, the tendency of validation accuracy is as same as test accuracy, even though the validation set is noisy. Therefore, the user can use the validation set to determine the best $\alpha$.

## H    THE INFLUENCE ON HARD EXAMPLES

If the loss of incorrectly-labeled examples is larger than the loss of hard but correctly-labeled examples (*e.g.*, the number of hard but correctly-labeled examples is more than the number of incorrectly-labeled examples), VRNL should not have a large negative impact on hard but correctly-labeled examples because it can still separate hard correctly-labeled examples from incorrectly-labeled examples.

If the loss of incorrectly-labeled examples equals the loss of hard but correctly-labeled examples (*i.e.*, these examples are entangled), it is hard to separate hard but correctly-labeled examples from

Table 6: Means and standard deviations (percentage) of classification accuracy. Results with "*" mean that they are the highest accuracy.

|  | MNIST | CIFAR-10 | CIFAR-100 |
|---|---|---|---|
| CE | **99.16 ± 0.03** | **92.23 ± 0.09** | 71.30 ± 0.16 |
| CE-VRNL | 99.13 ± 0.08 | 92.02 ± 0.17 | **71.40 ± 0.32** |

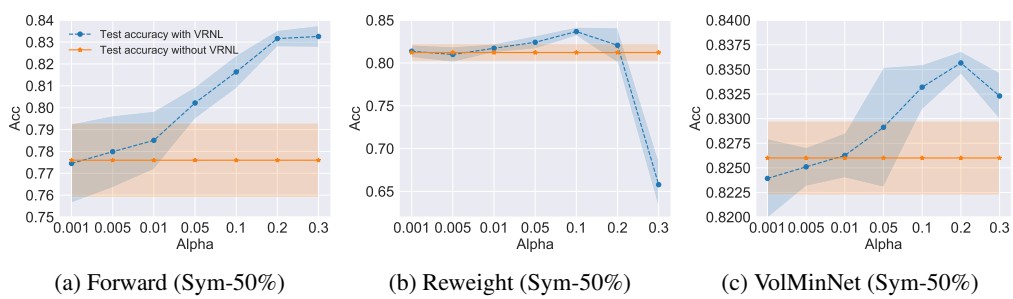

(a) Forward (Sym-50%)      (b) Reweight (Sym-50%)      (c) VolMinNet (Sym-50%)

Figure 8: Sensitivity analysis for $\alpha$.

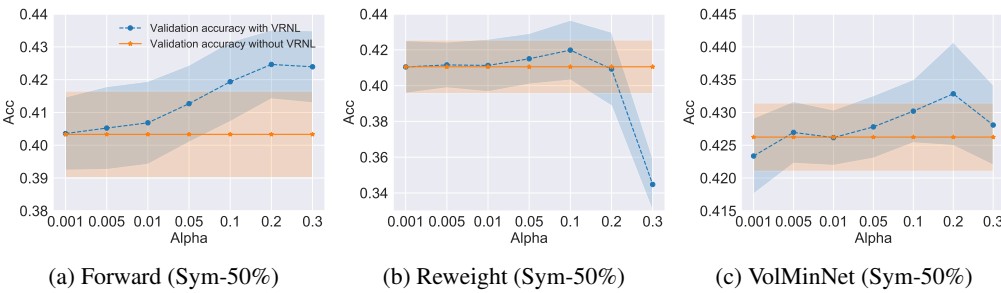

(a) Forward (Sym-50%)      (b) Reweight (Sym-50%)      (c) VolMinNet (Sym-50%)

Figure 9: Validation accuracy for different $\alpha$.

incorrectly-labeled examples. In such a case, all existing sample selection and reweighting methods would have the same problem.

We conduct an experiment on CIFAR-10 and found that the loss of hard but correctly-labeled examples is smaller than the loss of incorrectly-labeled examples. To find out hard examples, we train a ResNet-18 model on the clean dataset for 50 epochs and sort the cross-entropy loss of all training examples. The 30% examples with the largest loss are defined to be hard examples. Then we corrupt all training examples manually by using 50% symmetry-flipping noise and train a new ResNet-18 model using Forward-VRNL. The losses of hard but correctly-labeled examples are shown in Fig. 10. Hard but correctly-labeled examples and incorrectly-labeled examples can be separated very well, thus VRNL should not have a large negative impact on hard but correctly-labeled examples.

## I    EXPERIMENTS OF PROGRESSIVE EARLY STOPPING

We also conduct the experiment on Progressive Early Stopping with VRNL (PES-VRNL). The experiment results are shown and Tab. 7. The VRNL can still boost the performance of models.

## J    WARMING UP $\alpha$

The initial weights of VolMinNet are random (The weights of Forward and Reweight are acquired through early stopping which is also used to estimate transition matrix), the loss of instances with incorrect labels might not be larger than correct ones. Therefore, the influence of VRNL should

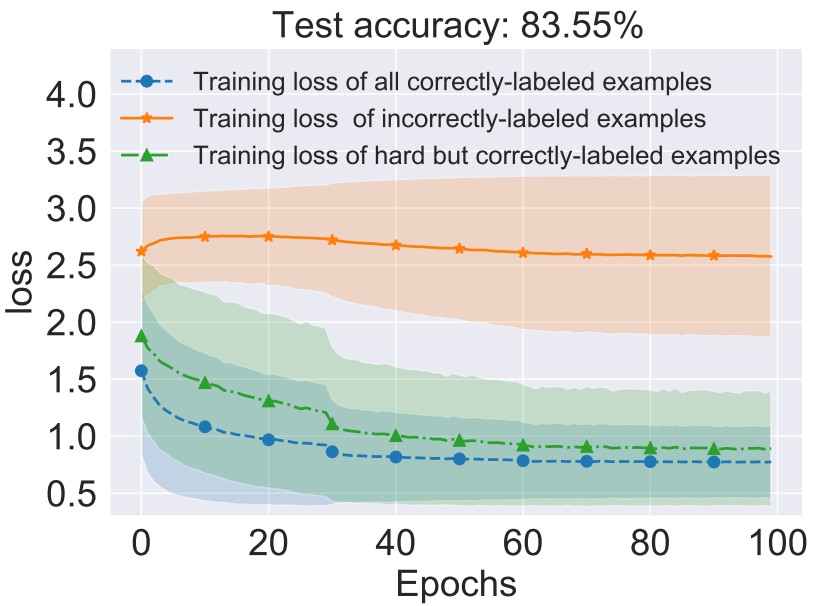

Figure 10: The influence of VRNL on hard examples.

Table 7: The test accuracy under Sym-50% noise on CIFAR-10.

|          | Sym-20%           | Sym-50%           |
| -------- | ----------------- | ----------------- |
| PES      | $92.57 \pm 0.22$  | $87.78 \pm 0.34$  |
| PES-VRNL | $\mathbf{92.66 \pm 0.09}$ | $\mathbf{87.85 \pm 0.17}$ |

not be large at first. We conduct the experiment on CIFAR-10 using VolMinNet, $\alpha$ is increased with time. Specifically, we increase $\alpha$ from 0 to 0.1 linearly every mini-batch. $\alpha$ peaks at 0.1 after 5 epochs. The experiment results are shown in Tab. 8. As can be seen in the experiments, the warming up strategy can increase the test accuracy compared with keeping $\alpha$ a constant.

Table 8: The test accuracy under Sym-50% noise.

|  | Sym-20% | Sym-50% |
| --- | --- | --- |
| VolMinNet | $89.27 \pm 0.30$ | $82.17 \pm 0.19$ |
| VolMinNet-VRNL (constant) | $89.42 \pm 0.12$ | $82.92 \pm 0.24$ |
| VolMinNet-VRNL (warming up) | $\mathbf{89.51 \pm 0.16}$ | $\mathbf{83.51 \pm 0.40}$ |

## K   THE LOSSES DURING TRAINING

To influence of VRNL on the losses distribution through training, we plot the training losses of the model with/without VRNL (correct/incorrect labeled examples together). The noise type is symmetry-flipping, and the noise rate is 0.5. The results are shown in Fig. 11.

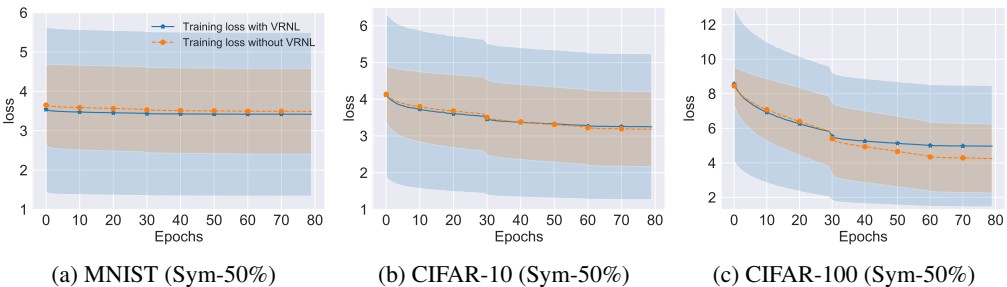

(a) MNIST (Sym-50%)          (b) CIFAR-10 (Sym-50%)          (c) CIFAR-100 (Sym-50%)

Figure 11: The training losses of models with VRNL and without VRNL.

The experiment results imply that when the model uses VRNL, the variance of training losses will increase.

## L   THE LOSSES AND VARIANCE OF LOSSES DURING TRAINING ON CLEAN AND NOISY DOMAIN

We conduct experiments, we train a Forward model on a clean dataset and a noisy dataset separately, the noise type is symmetry-flipping, and the noise rate is 0.5.

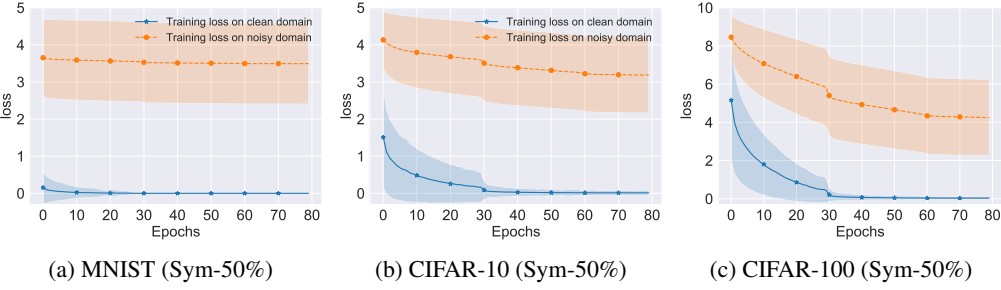

(a) MNIST (Sym-50%)          (b) CIFAR-10 (Sym-50%)          (c) CIFAR-100 (Sym-50%)

Figure 12: The training losses of models on clean and noisy domain.

## M   THE GRADIENTS OF REWEIGHT-VRNL

The full derivation of the gradients of Reweight-VRNL.

$$\nabla \hat{R}_{IR}(f_\theta) = \frac{1}{n} \sum_{i=1}^{n} \frac{\partial \hat{\beta}_i \ell_{CE}(f_\theta(x_i), \tilde{y}_i)}{\partial \theta}$$

$$- \alpha \left( \frac{1}{n} \sum_{i=1}^{n} 2\hat{\beta}_i \ell_{CE}(f_\theta(x_i), \tilde{y}_i) \frac{\partial \hat{\beta}_i \ell_{CE}(f_\theta(x_i), \tilde{y}_i)}{\partial \theta} \right.$$

$$\left. -2 \left( \frac{1}{n} \sum_{i=1}^{n} \hat{\beta}_i \ell_{CE}(f_\theta(x_i), \tilde{y}_i) \right) \left( \frac{1}{n} \sum_{i=1}^{n} \frac{\partial \hat{\beta}_i \ell_{CE}(f_\theta(x_i), \tilde{y}_i)}{\partial \theta} \right) \right),$$

where $\frac{1}{n} \sum_{i=1}^{n} \hat{\beta}_i \ell_{CE}(f_\theta(x_i), \tilde{y}_i)$ is a constant, we note it as $\hat{E}_{ir}$, therefore:

$$\nabla \hat{R}_{IR}(f_\theta) = \frac{1}{n} \sum_{i=1}^{n} \frac{\partial \hat{\beta}_i \ell_{CE}(f_\theta(x_i), \tilde{y}_i)}{\partial \theta}$$

$$- \alpha \left( \frac{1}{n} \sum_{i=1}^{n} 2\hat{\beta}_i \ell_{CE}(f_\theta(x_i), \tilde{y}_i) \frac{\partial \hat{\beta}_i \ell_{CE}(f_\theta(x_i), \tilde{y}_i)}{\partial \theta} \right.$$

$$\left. -2\hat{E}_{ir} \left( \frac{1}{n} \sum_{i=1}^{n} \frac{\partial \hat{\beta}_i \ell_{CE}(f_\theta(x_i), \tilde{y}_i)}{\partial \theta} \right) \right)$$

$$= \frac{1}{n} \sum_{i=1}^{n} \frac{\partial \hat{\beta}_i \ell_{CE}(f_\theta(x_i), \tilde{y}_i)}{\partial \theta}$$

$$- \alpha \left( \frac{1}{n} \sum_{i=1}^{n} 2\hat{\beta}_i \ell_{CE}(f_\theta(x_i), \tilde{y}_i) \frac{\partial \hat{\beta}_i \ell_{CE}(f_\theta(x_i), \tilde{y}_i)}{\partial \theta} \right.$$

$$\left. -\frac{1}{n} \sum_{i=1}^{n} 2\hat{E}_{ir} \frac{\partial \hat{\beta}_i \ell_{CE}(f_\theta(x_i), \tilde{y}_i)}{\partial \theta} \right)$$

$$= \frac{1}{n} \sum_{i=1}^{n} \left( \left( 1 - 2\alpha\hat{\beta}_i \ell_{CE}(f_\theta(x_i), \tilde{y}_i) + 2\alpha\hat{E}_{ir} \right) \frac{\partial \hat{\beta}_i \ell_{CE}(f_\theta(x_i), \tilde{y}_i)}{\partial \theta} \right).$$

