# OpenReview forum: "Do We Always Need to Penalize Variance of Losses for Learning with Label Noise?"
_ICLR.cc/2023/Conference — Submitted to ICLR 2023_

### Official Review · Reviewer_VFMQ · 2022-10-23

**Confidence:** 4
**Correctness:** 3
**Technical Novelty And Significance:** 3
**Empirical Novelty And Significance:** 3
**Recommendation:** 6

**Clarity, Quality, Novelty And Reproducibility:**

* Clarity: the paper is well-written with a clear exposition
* Novelty: the proposed approach is contrary to traditional wisdom and the authors make a compelling case for its benefits.
* Quality: the empirical results provide support for the method when used with state-of-the-art label-noise learning approaches.
* Reproducibility: the hyperparameters and the configurations used for the experiments are outlined in the results section (Sec. 4).


**Strength And Weaknesses:**

## Strengths
* Learning with noisy labels is a topic of high importance and interest to the wider ML community.
* The proposed technique is novel and contrary to the traditional advice of penalizing the variance. The authors make a compelling case for the benefit of rewarding the variance in terms of attenuating the influence of noisy labels on the model during training.
* The authors present extensive empirical evidence on a variety of scenarios that demonstrate the improved effectiveness of the method in boosting the performance of existing label-noise learning approaches.
* Sensitivity analysis of the method’s hyperparameter is provided in the supplementary that shows that it can be estimated using a validation data set

## Weaknesses
* Even though there are experiments with VRNL augmenting prior label-noise learning approaches, it would have been very compelling to include results with VRNL alone, without knowledge or reasoning about the transition matrix. For instance, a performance comparison of VRNL vs. no VRNL on a simple CIFAR10 example where, e.g., 10% of the labels are artificially made to be incorrect, would go a long way.
* The method requires tuning of the hyperparameter $\alpha$ and a wide range of values for $\alpha$ is used for the various experiments on different scenarios in Sec. 4. From the plots in the appendix (Fig. 7), the method’s performance is highly sensitive to the precise value of $\alpha$. The authors did address that a validation data set can be used in a reliable way for CIFAR10, however, it is not clear whether this is always the case.


**Summary Of The Paper:**

This paper makes the seemingly counterintuitive claim that the variance of losses should be rewarded, rather than penalized, in problems involving learning with noisy labels. Mathematically, the authors show that rewarding the variance of losses leads to parameter updates so that if the loss of the example is smaller than the average loss over the examples, the magnitude of the update is magnified, and vice-versa. This leads to an intuitive safeguard against the influence of points with noisy labels, which tend to have high loss. The authors present extensive empirical results that show that this type of variance regularization boosts the performance of existing methods for learning with noisy labels.

**Summary Of The Review:**

This paper addresses a problem that is highly relevant to the ML community. It challenges the traditional wisdom that variance of losses should be penalized, and it instead makes a compelling argument for rewarding the variance. Even though a more direct comparison of VRNL would go a long way (see above), the authors present compelling empirical evidence on the benefits of the proposed method when used in conjunction with prior methods. In light of these considerations, I recommend acceptance.

---

> ### Author Response · Authors · 2022-11-15
> **Response to Reviewer VFMQ**
>
> **Q1: It would have been very compelling to include results with VRNL alone.**
>
> A1:  We have added the results of standard cross-entropy loss with VRNL in our main paper. It shows that VRNL can improve the performance of standard cross-entropy loss under different types of label noise.
>
> **Symmetry-flipping**
>
> |         |        MNIST         |        MNIST         |       CIFAR-10       |       CIFAR-10       |      CIFAR-100       |      CIFAR-100       |
> | ------- | :------------------: | :------------------: | :------------------: | :------------------: | :------------------: | :------------------: |
> |         |    noise rate 0.2    |    noise rate 0.5    |    noise rate 0.2    |    noise rate 0.5    |    noise rate 0.2    |    noise rate 0.5    |
> | CE      |   98.65 $\pm$ 0.05   |   97.94 $\pm$ 0.17   |   86.86 $\pm$ 0.26   |   78.93 $\pm$ 0.47   |   60.15 $\pm$ 0.46   |   45.66 $\pm$ 0.68   |
> | CE-VRNL | **98.71 $\pm$ 0.06** | **97.96 $\pm$ 0.06** | **87.49 $\pm$ 0.25** | **79.31 $\pm$ 0.52** | **60.38 $\pm$ 0.6**1 | **47.79 $\pm$ 0.52** |
>
> **Asymmetry-flipping**
>
> |         |        MNIST         |        MNIST         |       CIFAR-10       |       CIFAR-10       |      CIFAR-100       |      CIFAR-100       |
> | ------- | :------------------: | :------------------: | :------------------: | :------------------: | :------------------: | :------------------: |
> |         |    noise rate 0.2    |    noise rate 0.5    |    noise rate 0.2    |    noise rate 0.5    |    noise rate 0.2    |    noise rate 0.5    |
> | CE      |   98.76 $\pm$ 0.07   |   97.91 $\pm$ 0.23   |   87.31 $\pm$ 0.32   |   79.47 $\pm$ 0.47   |   59.83 $\pm$ 0.69   |   45.08 $\pm$ 0.71   |
> | CE-VRNL | **98.77 $\pm$ 0.12** | **97.98 $\pm$ 0.20** | **87.38 $\pm$ 0.31** | **79.61 $\pm$ 0.40** | **60.69 $\pm$ 0.51** | **45.44 $\pm$ 0.10** |
>
> **Pair-flipping**
>
> |         |        MNIST         |        MNIST         |       CIFAR-10       |       CIFAR-10       |      CIFAR-100       |      CIFAR-100       |
> | ------- | :------------------: | :------------------: | :------------------: | :------------------: | :------------------: | :------------------: |
> |         |    noise rate 0.2    |    noise rate 0.5    |    noise rate 0.2    |    noise rate 0.5    |    noise rate 0.2    |    noise rate 0.5    |
> | CE      |   98.71 $\pm$ 0.08   |   83.49 $\pm$ 3.77   |   88.63 $\pm$ 0.26   |   66.32 $\pm$ 2.44   | **61.04 $\pm$ 0.31** |   39.78 $\pm$ 0.30   |
> | CE-VRNL | **98.80 $\pm$ 0.10** | **84.00 $\pm$ 3.65** | **88.71 $\pm$ 0.30** | **67.71 $\pm$ 2.04** |   61.00 $\pm$ 0.32   | **39.91 $\pm$ 0.20** |
>
> **Q2: It is not clear whether the noisy validation data set can be always used to select $\alpha$.**
>
> A2: Yes, the noisy validation data set can be always used to select the hyper-parameter.
> Specifically, our method is mainly designed for statistically consistent methods, which can employ a noisy validation dataset to choose hyperparameters [1].  Although VRNL adds a regularizer to the statistically consistent objective with a small trade-off parameter, it has a small influence on consistency. A noisy validation dataset could work well. We have also empirically validated this by visualizing both test accuracy and validation accuracy for each training epoch. It shows that the tendency of test accuracy follows the tendency of validation accuracy. Therefore, the noisy validation set can be used to select the best $\alpha$.
>
> **References**
>
> [1] Patrini, Giorgio, et al. "Making deep neural networks robust to label noise: A loss correction approach." Proceedings of the IEEE conference on computer vision and pattern recognition. 2017.

---

> > ### Comment · Reviewer_VFMQ · 2022-12-11
> > **Thank you**
> >
> > Thank you for your response and the additional experiments.
> >
> > I read the reviews and all of the responses to the paper. We also had an internal discussion with the other reviewers. Even though I think that the paper still has some merit, the internal discussion and the other reviewers' comments have made me more clearly aware of the concerns over the statistical significance of the method's empirical benefit (including the new results comparing CE and CE-VRNL) and the complexity of the hyperparameter $\alpha$ and the role that it plays in providing increased modeling capacity. The reviewers also shared my initial concerns regarding the sensitivity of the method to $\alpha$.
> >
> > In light of the above, I have decided to lower my score.

---

> > > ### Author Response · Authors · 2022-12-11
> > > **Thank you**
> > >
> > > Dear Reviewer VFMQ,
> > >
> > > Thanks for your kind feedback. We appreciate it very much and will try our best to improve the paper in the future.
> > >
> > > Best wishes,
> > > Authors

---

### Official Review · Reviewer_zTmG · 2022-10-24

**Confidence:** 4
**Correctness:** 3
**Technical Novelty And Significance:** 2
**Empirical Novelty And Significance:** 2
**Recommendation:** 5

**Clarity, Quality, Novelty And Reproducibility:**

Clarity:
The paper is mostly clear, however the experiments are scattered between the main body and appendix, which makes the reading harder.

Quality:
While the empirical validation is relatively thorough, the paper's writing quality can be improved.

Novelty:
The idea of exploiting early memorization of the network is not novel. Adding loss variance as a regularization to achieve better resilience to noise seems to be novel (to the best of my knowledge).

Reproducibility:
The authors provide code as a supplementary material.

**Details Of Ethics Concerns:**

No concerns

**Strength And Weaknesses:**

Strength:
The overall idea of adjusting the weights, such that noisy samples influence weight update less, sounds like a good idea. The approach suggested to achieve that with addition of variance component to the loss is relatively simple (can be applicable as the authors show to several techniques) and yet relatively effective (improved results compared to the original method -- Tables 1 & 2).

Weakness:
The paper mostly proves the effectiveness of the approach empirically, improving the results by some margin in case of the synthetic noise. The improvement on real noise ( Clothing1M  dataset) is relatively minor (Table 2 -- <0.5% improvement) and since there is no real theoretical proof that the suggested approach should work and under what conditions, the paper overall is not very convincing (in other worlds, that would not be my choice of an approach when dealing with noisy labels).
 In the absence of theory and given that the paper is not aiming to provide a SOTA technique, but discuss the loss variance penalization / encouragement, reorganization of the paper, e.g. :
(1) bringing results from the appendix to the main body (e.g., uniting Table 1 and Table 4&5); (2) writing a paragraph with concrete conclusions and answer to the question posed in the title; (3)  clear conclusion on what types of noise, datasets  (balanced / not balanced) benefit from the approach, can help to strengthen the empirical contribution of the paper.

**Summary Of The Paper:**

The paper aimed to explore if the variance of losses should be always penalized when learning with noisy labels. The paper suggests adding a variance regularization to the loss, which will encourage the variance of losses. Consequently, when calculating the gradients of the proposed loss, if the loss of an example  is smaller than the expectation of the losses (which the authors assume is true in the case of examples with clean labels), the weight associated with the respective gradient will be increased and the example contributes more to the update of the parameter. If the loss of the example is larger than the expectation of the losses, the weight associated with its gradient is small and the example contributes less to the update of parameter. The assumption is that the latest will happen for noisy examples, because the network tends first to learn from clean examples and increasing the variance of losses would boost this memorization effect and reduce the harmfulness of incorrect labels.
The authors explain how to add the suggested regularization to several recent methods and show improvement.

**Summary Of The Review:**

I have mixed feelings about this paper. I would suggest the authors reorganize the paper in a way that it provides some insights beyond just illustrating improvements to some methods (and providing the implementation details).
1) The authors ask the question -- do we always need to finalize variances?  -- I could not find a clear to this question. It seems that the authors actually the opposite:  that the variances should be encouraged. So is the answer - always encourage losses?
2) I am wondering why the authors did not provide the evaluation results of their method on WebVision dataset. Given relatively minor improvement on Clothing1M (< 0.5%), how will the results look like for WebVision?
3) In my opinion, it's not clear why the authors push the results with little noise and strong noise to the appendix. It will be interesting to see those together with the results of Table 1 and maybe as a graph that shows the improvements (deltas)  as a function of noise.

I will definitely reconsider my rating if the paper is reorganized and the main learning are clearly stated.

---

> ### Author Response · Authors · 2022-11-15
> **Response to Reviewer zTmG**
>
> **Q1: The improvement on real noise (Clothing1M dataset) is relatively minor, how will the results look like for WebVision?**
>
> A1: We also conduct the experiment on WebVision and follow the same settings of DivideMix. Our method improves the performance of existing statistically consistent methods.
>
> |              | Forward   | Reweight  | VolMinNet |
> | ------------ | --------- | --------- | --------- |
> | Without VRNL | 33.40     | 46.28     | 72.44     |
> | With VRNL    | **35.00** | **48.44** | **72.92** |
>
> **Q2: Under what conditions, VRNL works? Should the loss be always increased for learning with noisy labels? Clear conclusion on what types of noise, datasets (balanced / not balanced) benefit from the approach.**
>
> A2:  VRNL can dramatically improve the performance of statistically consistent methods when the training sample size is limited. Empirically, it shows that even on Clothing1M dataset with $1$ million training data, VRNL still improves the performance of statistically consistent methods.
>
> We will explain the reason that how VRNL helps existing statistically consistent methods improve their performance as follows.
>
>
> For statistically consistent learning algorithms, a label-noise transition matrix is usually required to correct the loss function. The high-level idea is that, the expected risk of the corrected loss function with respect to the noisy data is proportional to the expected risk of the original loss with respect to the clean data. Intuitively, when the training data size is sufficiently large, the noisy data can be "corrected" into clean data with the help of the transition matrix.  The statistically consistent learning algorithms therefore minimize the average corrected loss, aiming to make each example's loss small and have a small variance of losses. In other words, the statistically consistent algorithms should fit noisy data with the help of the transition matrix asymptotically.
>
> However, when the training sample size is limited, the noisy data cannot be "corrected" to clean data. We argue that we should not let the model fit the whole data. An extreme case is that if we do not do any "correction", we should let the model only fit the correctly labeled examples but not the incorrectly labeled examples because we do not want the model to fit noise. Based on our argument, for statistically consistent algorithms, when the training sample size is finite, we should not let the variance of the corrected loss be small. In other words, we do not want the model to fit the not well "corrected" data which contains noise.
>
> It is also worth mentioning that, increasing the variance of loss may not work in some extreme cases. For example, the label noise is specifically designed such that the variance of loss on noisy data is smaller than or equal to the variance on clean data. Then increasing the variance of loss may have a negative influence. However, in this case, all other sample selection or reweighting methods that implicitly increase the variance have the same problem.

---

> > ### Author Response · Authors · 2022-11-15
> > **Response to Reviewer zTmG (Continue)**
> >
> > **Q3: Bringing results from the appendix to the main body.**
> >
> > A3: We have merged the results of CE and CE-VRNL with the results of Table 1. The results show that our method can generally improve  backbone methods on different levels of noise. We have also drawn a graph to illustrate the change in the accuracy of VRNL with the increase of noise in the modified version.
> >
> > **Symmetry-flipping**
> >
> > |                |        MNIST         |        MNIST         |       CIFAR-10       |       CIFAR-10       |      CIFAR-100       |      CIFAR-100       |
> > | -------------- | :------------------: | :------------------: | :------------------: | :------------------: | :------------------: | :------------------: |
> > |                |    noise rate 0.2    |    noise rate 0.5    |    noise rate 0.2    |    noise rate 0.5    |    noise rate 0.2    |    noise rate 0.5    |
> > | CE             |   98.65 $\pm$ 0.05   |   97.94 $\pm$ 0.17   |   86.86 $\pm$ 0.26   |   78.93 $\pm$ 0.47   |   60.15 $\pm$ 0.46   |   45.66 $\pm$ 0.68   |
> > | CE-VRNL        | **98.71 $\pm$ 0.06** | **97.96 $\pm$ 0.06** | **87.49 $\pm$ 0.25** | **79.31 $\pm$ 0.52** | **60.38 $\pm$ 0.6**1 | **47.79 $\pm$ 0.52** |
> > | Forward        |   97.47 $\pm$ 0.15   |   97.93 $\pm$ 0.22   |   87.29 $\pm$ 0.63   |   77.58 $\pm$ 1.05   |   59.71 $\pm$ 0.40   |   44.53 $\pm$ 1.11   |
> > | Forward-VRNL   | **98.89 $\pm$ 0.04** | **98.14 $\pm$ 0.27** | **89.81 $\pm$ 0.29** | **81.16 $\pm$ 0.55** | **68.19 $\pm$ 0.31** | **54.10 $\pm$ 1.2**  |
> > | Reweight       |   98.20 $\pm$ 0.24   |   97.93 $\pm$ 0.20   |   88.42 $\pm$ 0.18   |   82.13 $\pm$ 0.56   |   60.52 $\pm$ 0.52   |   47.69 $\pm$ 0.78   |
> > | Reweight-VRNL  | **98.61 $\pm$ 0.21** | **98.27 $\pm$ 0.15** | **89.68 $\pm$ 0.24** | **83.99 $\pm$ 0.28** | **66.52 $\pm$ 0.25** | **50.26 $\pm$ 0.14** |
> > | VolMinNet      |   98.66 $\pm$ 0.14   |   97.83 $\pm$ 0.15   |   89.27 $\pm$ 0.30   |   82.17 $\pm$ 0.19   |   65.65 $\pm$ 0.62   |   54.40 $\pm$ 0.62   |
> > | VolMinNet-VRNL | **98.78 $\pm$ 0.08** | **97.93 $\pm$ 0.20** | **89.42 $\pm$ 0.12** | **82.92 $\pm$ 0.24** | **66.40 $\pm$ 0.66** | **55.94 $\pm$ 0.64** |
> >
> > **Asymmetry-flipping**
> >
> > |                |        MNIST         |        MNIST         |       CIFAR-10       |       CIFAR-10       |      CIFAR-100       |      CIFAR-100       |
> > | -------------- | :------------------: | :------------------: | :------------------: | :------------------: | :------------------: | :------------------: |
> > |                |    noise rate 0.2    |    noise rate 0.5    |    noise rate 0.2    |    noise rate 0.5    |    noise rate 0.2    |    noise rate 0.5    |
> > | CE             |   98.76 $\pm$ 0.07   |   97.91 $\pm$ 0.23   |   87.31 $\pm$ 0.32   |   79.47 $\pm$ 0.47   |   59.83 $\pm$ 0.69   |   45.08 $\pm$ 0.71   |
> > | CE-VRNL        | **98.77 $\pm$ 0.12** | **97.98 $\pm$ 0.20** | **87.38 $\pm$ 0.31** | **79.61 $\pm$ 0.40** | **60.69 $\pm$ 0.51** | **45.44 $\pm$ 0.10** |
> > | Forward        |   98.42 $\pm$ 0.06   |   97.92 $\pm$ 0.04   |   87.70 $\pm$ 0.29   |   79.25 $\pm$ 1.61   |   60.24 $\pm$ 0.42   |   43.39 $\pm$ 1.15   |
> > | Forward-VRNL   | **98.92 $\pm$ 0.06** | **98.13 $\pm$ 0.21** | **89.98 $\pm$ 0.11** | **82.35 $\pm$ 0.88** | **67.89 $\pm$ 0.30** | **53.67 $\pm$ 0.52** |
> > | Reweight       |   98.50 $\pm$ 0.07   |   98.09 $\pm$ 0.08   |   88.55 $\pm$ 0.32   |   82.72 $\pm$ 0.38   |   60.81 $\pm$ 0.70   |   46.36 $\pm$ 0.18   |
> > | Reweight-VRNL  | **98.77 $\pm$ 0.21** | **98.10 $\pm$ 0.16** | **89.80 $\pm$ 0.11** | **84.20 $\pm$ 0.25** | **66.62 $\pm$ 0.45** | **49.71 $\pm$ 0.64** |
> > | VolMinNet      |   98.62 $\pm$ 0.10   |   98.03 $\pm$ 0.12   |   89.50 $\pm$ 0.18   |   83.15 $\pm$ 0.56   |   66.02 $\pm$ 0.73   |   55.17 $\pm$ 0.46   |
> > | VolMinNet-VRNL | **98.76 $\pm$ 0.13** | **98.08 $\pm$ 0.08** | **89.64 $\pm$ 0.19** | **83.65 $\pm$ 0.32** | **66.24 $\pm$ 0.95** | **55.85 $\pm$ 0.73** |

---

> > > ### Author Response · Authors · 2022-11-15
> > > **Response to Reviewer zTmG (Continue)**
> > >
> > > **Pair-flipping**
> > >
> > > |                |        MNIST         |        MNIST         |       CIFAR-10       |       CIFAR-10       |      CIFAR-100       |      CIFAR-100       |
> > > | -------------- | :------------------: | :------------------: | :------------------: | :------------------: | :------------------: | :------------------: |
> > > |                |    noise rate 0.2    |    noise rate 0.5    |    noise rate 0.2    |    noise rate 0.5    |    noise rate 0.2    |    noise rate 0.5    |
> > > | CE             |   98.71 $\pm$ 0.08   |   83.49 $\pm$ 3.77   |   88.63 $\pm$ 0.26   |   66.32 $\pm$ 2.44   | **61.04 $\pm$ 0.31** |   39.78 $\pm$ 0.30   |
> > > | CE-VRNL        | **98.80 $\pm$ 0.10** | **84.00 $\pm$ 3.65** | **88.71 $\pm$ 0.30** | **67.71 $\pm$ 2.04** |   61.00 $\pm$ 0.32   | **39.91 $\pm$ 0.20** |
> > > | Forward        |   98.85 $\pm$ 0.09   |   96.45 $\pm$ 4.03   | **90.88 $\pm$ 0.23** |   83.27 $\pm$ 9.47   |   62.54 $\pm$ 0.42   |   41.96 $\pm$ 1.45   |
> > > | Forward-VRNL   | **98.88 $\pm$ 0.08** | **96.55 $\pm$ 3.88** | **90.88 $\pm$ 0.29** | **83.54 $\pm$ 9.29** | **62.78 $\pm$ 0.32** | **42.29 $\pm$ 1.23** |
> > > | Reweight       |   98.64 $\pm$ 0.07   |   95.52 $\pm$ 3.58   |   89.68 $\pm$ 0.30   |   76.03 $\pm$ 5.02   |   61.35 $\pm$ 0.66   |   40.10 $\pm$ 0.46   |
> > > | Reweight-VRNL  | **98.68 $\pm$ 0.14** | **95.97 $\pm$ 3.52** | **89.83 $\pm$ 0.30** | **76.75 $\pm$ 6.15** | **61.37 $\pm$ 0.42** | **40.30 $\pm$ 0.57** |
> > > | VolMinNet      | **99.05 $\pm$ 0.05** |   99.08 $\pm$ 0.06   |   90.73 $\pm$ 0.23   |   88.47 $\pm$ 0.61   |   69.96 $\pm$ 1.18   |   61.85 $\pm$ 1.41   |
> > > | VolMinNet-VRNL |   99.02 $\pm$ 0.08   | **99.10 $\pm$ 0.08** | **90.86 $\pm$ 0.27** | **88.77 $\pm$ 0.51** | **70.18 $\pm$ 0.50** | **63.38 $\pm$ 1.72** |
> > >
> > > ##

---

> > > > ### Comment · Reviewer_zTmG · 2022-11-29
> > > > **Response to the authors**
> > > >
> > > > Thank you for your responses.
> > > > I read other reviewers' comments, your responses and also looked at the revised paper. I still feel that the paper could be revised to accomodate your responses within the main text.
> > > > At this point I'm not inclined to change my decision, but will participate in a discussion if such is initiated by other reviewers.

---

> > > > > ### Author Response · Authors · 2022-11-29
> > > > > **Thanks**
> > > > >
> > > > > Dear Reviewer zTmG,
> > > > >
> > > > > Thanks for your response and suggestion. The rolling discussion phase is to make sure the rebuttals are satisfactory and proper. In the end, we definitely will incorporate the rebuttal into the main paper and supplementary material in the final version. If you have any remaining concerns or questions, please kindly let us know.
> > > > >
> > > > > Best wishes,
> > > > > Authors

---

> ### Author Response · Authors · 2022-11-30
> **Further discussion**
>
> Dear Reviewer zTmG,
>
> If you can tell us the detailed contents in our responses that should be added to the main text, we will be grateful. In our opinion, we should add more detailed analyses of the relationship between VRNL and existing selection/reweighting methods to the main paper; we should merge the results of webvision into Sec. 4.1; We should briefly state the reason why validation sets can be used to select a suitable $\alpha$ in Sec. 3.1. If we have any misunderstandings or omissions, please let us know. We should discuss them to make them clear during this phase. I advise that we can discuss this with other reviewers in a new pose.
>
> Best wishes, Authors

---

### Official Review · Reviewer_JfaU · 2022-11-01

**Confidence:** 3
**Correctness:** 3
**Technical Novelty And Significance:** 3
**Empirical Novelty And Significance:** 2
**Recommendation:** 5

**Clarity, Quality, Novelty And Reproducibility:**

*Minor*

- Fig 1: what dataset is the results reported on
- page 4: “noise examples” -> noisy
- page 4: “when work” -> working
- page 5 “log determinate” -> determinant

**Strength And Weaknesses:**

- the paper proposes a simple idea for robustness to label noise

- some results are encouraging compared to the chosen baselines.

- the method seems original in that it enforces high variance on the training data loss. However, quite a few methods that apply some form of selection/reweighting/distribution on the individual sample loss functions can be seen as increasing the variance of training losses. It is important to make such discussion in the paper and highlight the differences of the paper's proposal to such prior works both formally and empirically.

- since the paper's claim on increasing variance for noise robustness is general, either the main experiments should be done on standard loss functions such as CE or the claims and theoretical motivation should be adjusted specifically to the case of noise transition matrices.

- due to the simplicity of the approach (increasing loss variance) it is natural to assume complications might arise from the confusion of hard but correctly-labelled examples with noisily-labelled examples. Therefore, extensive discussions and experiments are required as to how such variance encouragement is theoretically or empirically unharmful for the difficult examples (e.g., in a scenario where the data is clean but quite complex to model).

- different $\alpha$s are used for different setups, what is the sensitivity of the method to $\alpha$ (in each/some of those setups)? How is  optimized for each setup? Also, it seems like too-early or too-high enforcement of large variances can be detrimental to fitting the cleanly labelled data. Does the method require heavy hyperparameter tuning? If not, how robust is it to hyperparameters? Why is it robust?



**Summary Of The Paper:**

- the paper proposes a simple idea for robustness to label noise. The idea is to increase the variance of the loss function across training samples. A high variance could generally make sense since ideally the loss remains high for noisy-labelled samples during training while on the other hand the loss for correctly-labelled data is gradually decreased.

- it further derives a form for the derivative of the proposed total loss w.r.t. the model parameters where the proposed loss can be seen as a reweighting of the gradient induced by individual training samples. It proportionally upweights the gradient of those samples with lower losses than the average loss and downweights the gradient of the samples with higher losses than the average loss.

- it applies the method on top of three noise robustness techniques. Two of them are based on noisy label transition matrices, namely Forward and VolMinNet as well as a loss reweighting method.

- the experiments show consistent improvement (although sometimes small) of the modified version of those three methods on different datasets, MNIST, CIFAR, Clothing1M) and noise types and rates (real, synthetic, symmetric, and asymmetric, 30% and 50%).

- some additional studies are done to shed light on the inner workings of the proposed method. Interestingly, it is shown that the proposed method is more robust to erroneous estimation of the noisy label transition matrix.

**Summary Of The Review:**

The paper proposes a simple change of the loss function that encourages variance of the loss across training samples and shows this can bring robustness to training label noise in some scenarios. However, there are some important concerns with the empirical evidence that has been provided in support of the paper's proposal.

---

> ### Author Response · Authors · 2022-11-15
> **Response to Reviewer JfaU**
>
> **Q1: Some selection/reweighting/distribution on the individual sample loss functions can be seen as increasing the variance of training losses. Highlight the differences between the proposed method and existing sample selection or reweighting methods.**
>
> A1: Thank you for the insightful question. VRNL explicitly increases the variance of losses on noisy training examples to prevent models from overfitting label noise.
>
> For statistically consistent learning algorithms, a label-noise transition matrix is usually required to correct the loss function. The high-level idea is that, the expected risk of the corrected loss function with respect to the noisy data is proportional to the expected risk of the original loss with respect to the clean data. Intuitively, when the training data size is sufficiently large, the noisy data can be "corrected" into clean data with the help of the transition matrix.  The statistically consistent learning algorithms therefore minimize the average corrected loss, aiming to make each example's loss small and have a small variance of losses. In other words, the statistically consistent algorithms should fit noisy data with the help of the transition matrix asymptotically.
>
> However, when the training sample size is limited, the noisy data cannot be "corrected" to clean data. We argue that we should not let the model fit the whole data. An extreme case is that if we do not do any "correction", we should let the model only fit the correctly labeled examples but not the incorrectly labeled examples because we do not want the model to fit noise. Based on our argument, for statistically consistent algorithms, when the training sample size is finite, we should not let the variance of the corrected loss be small (in other words, we do not want the model to fit the not well "corrected" data which contains noise).
>
> Intuitively, to prevent the model from overfitting noise, without loss correction, the loss value of the original loss function with respect to correctly labeled data should be small while the loss value with respect to incorrectly labeled data should be large. The variance of the losses with respect to overall dataset is large. For example, existing sample selection methods first select confident examples based on the small-loss trick. Then the model only trained on the confident examples but not unconfident examples. As a result, the loss of confident examples decreases while the loss of unconfident examples is usually large because the model does not learn unconfident examples, which increases the variance of losses on the training set implicitly.
> Similarly, the reweighting techniques will increase the variance of the original losses to prevent the model from overfitting noise.
>
> Now we can conclude the differences between the proposed methods and the existing ones:
>
> 1. We explicitly increase the variance with a clear objective
>    that prevents the model from overfitting noise, while the existing methods implicitly do so.
> 2. We mainly target the existing statistically consistent algorithms, i.e., increase the variance of corrected losses when the noisy training sample is finite. The existing method implicitly increases the variance of the original loss. Note that our idea to increase the variance of loss to prevent the model from overfitting label noise is general and can be used to better understand existing label-noise robust algorithms.
> 3. Another major difference is that most existing methods have to adjust the hyperparameter on a small set of clean data. For instance, the sample reweighting method [1] requires a clean validation set to learn the weight $w$. On the contrary, our method can use a noisy validation dataset for validation. Specifically, our method is mainly designed for statistically consistent methods, which can employ a noisy validation dataset to choose hyperparameters [2]. Although VRNL adds a regularizer to the statistically consistent objective with a small trade-off parameter, it has a small influence on consistency. A noisy validation dataset could work well.

---

> > ### Author Response · Authors · 2022-11-15
> > **Response to Reviewer JfaU (Continue)**
> >
> > **Q2: Either the main experiments should be done on standard loss functions such as CE or the claims and theoretical motivation should be adjusted specifically to the case of noise transition matrices.**
> >
> > A2: The main claims and theoretical motivation in our paper are mainly based on statistically consistent methods (i.e., the methods based on noise transition matrices) but not mainly designed for standard CE loss. Our major claim is that encouraging the loss variance can prevent statistically consistent methods from overfitting  noisy labels when the training sample size is limited.  As the reason we mentioned in A1, when the training data size is sufficiently large, the noisy data can be "corrected" into clean data with the help of the transition matrix.
> > The statistically consistent learning algorithms therefore minimize the average corrected loss, aiming to make each loss small and have a small variance of losses. In other words, the statistically consistent algorithms should fit noisy data with the help of the transition matrix asymptotically. However, when the training sample size is finite, the noisy data cannot be "corrected" to clean data. We argue that we should not let the model fit the whole data. Then we should not let the variance of the corrected loss be small because we do not want the model to fit the not well "corrected" data which contains noise.
> >
> > We have also shown that our method can also work with a standard CE-loss function to alleviate the overfitting problem.  For the standard CE-loss function,  the noisy data has not be "corrected" at all, we should let the model only fit the correctly labeled examples but not the incorrectly labeled examples because we do not want the model to fit noise. Then we should not let the variance of the corrected loss be small, which leads the model to fit incorrectly labeled examples. In this case, VRNL should be applied because it explicitly increases the variance of loss, which prevents the model from fitting incorrectly labeled examples. We have added the experiment of standard cross-entropy loss in the main paper of the revised version. The results show that VRNL improves the performance of the standard CE-loss function under different types of noise.
> >
> > **Symmetry-flipping**
> >
> > |                |        MNIST         |        MNIST         |       CIFAR-10       |       CIFAR-10       |      CIFAR-100       |      CIFAR-100       |
> > | -------------- | :------------------: | :------------------: | :------------------: | :------------------: | :------------------: | :------------------: |
> > |                |    noise rate 0.2    |    noise rate 0.5    |    noise rate 0.2    |    noise rate 0.5    |    noise rate 0.2    |    noise rate 0.5    |
> > | CE             |   98.65 $\pm$ 0.05   |   97.94 $\pm$ 0.17   |   86.86 $\pm$ 0.26   |   78.93 $\pm$ 0.47   |   60.15 $\pm$ 0.46   |   45.66 $\pm$ 0.68   |
> > | CE-VRNL        | **98.71 $\pm$ 0.06** | **97.96 $\pm$ 0.06** | **87.49 $\pm$ 0.25** | **79.31 $\pm$ 0.52** | **60.38 $\pm$ 0.6**1 | **47.79 $\pm$ 0.52** |
> > | Forward        |   97.47 $\pm$ 0.15   |   97.93 $\pm$ 0.22   |   87.29 $\pm$ 0.63   |   77.58 $\pm$ 1.05   |   59.71 $\pm$ 0.40   |   44.53 $\pm$ 1.11   |
> > | Forward-VRNL   | **98.89 $\pm$ 0.04** | **98.14 $\pm$ 0.27** | **89.81 $\pm$ 0.29** | **81.16 $\pm$ 0.55** | **68.19 $\pm$ 0.31** | **54.10 $\pm$ 1.2**  |
> > | Reweight       |   98.20 $\pm$ 0.24   |   97.93 $\pm$ 0.20   |   88.42 $\pm$ 0.18   |   82.13 $\pm$ 0.56   |   60.52 $\pm$ 0.52   |   47.69 $\pm$ 0.78   |
> > | Reweight-VRNL  | **98.61 $\pm$ 0.21** | **98.27 $\pm$ 0.15** | **89.68 $\pm$ 0.24** | **83.99 $\pm$ 0.28** | **66.52 $\pm$ 0.25** | **50.26 $\pm$ 0.14** |
> > | VolMinNet      |   98.66 $\pm$ 0.14   |   97.83 $\pm$ 0.15   |   89.27 $\pm$ 0.30   |   82.17 $\pm$ 0.19   |   65.65 $\pm$ 0.62   |   54.40 $\pm$ 0.62   |
> > | VolMinNet-VRNL | **98.78 $\pm$ 0.08** | **97.93 $\pm$ 0.20** | **89.42 $\pm$ 0.12** | **82.92 $\pm$ 0.24** | **66.40 $\pm$ 0.66** | **55.94 $\pm$ 0.64** |

---

> > > ### Author Response · Authors · 2022-11-15
> > > **Response to Reviewer JfaU (Continue)**
> > >
> > > **Asymmetry-flipping**
> > >
> > > |                |        MNIST         |        MNIST         |       CIFAR-10       |       CIFAR-10       |      CIFAR-100       |      CIFAR-100       |
> > > | -------------- | :------------------: | :------------------: | :------------------: | :------------------: | :------------------: | :------------------: |
> > > |                |    noise rate 0.2    |    noise rate 0.5    |    noise rate 0.2    |    noise rate 0.5    |    noise rate 0.2    |    noise rate 0.5    |
> > > | CE             |   98.76 $\pm$ 0.07   |   97.91 $\pm$ 0.23   |   87.31 $\pm$ 0.32   |   79.47 $\pm$ 0.47   |   59.83 $\pm$ 0.69   |   45.08 $\pm$ 0.71   |
> > > | CE-VRNL        | **98.77 $\pm$ 0.12** | **97.98 $\pm$ 0.20** | **87.38 $\pm$ 0.31** | **79.61 $\pm$ 0.40** | **60.69 $\pm$ 0.51** | **45.44 $\pm$ 0.10** |
> > > | Forward        |   98.42 $\pm$ 0.06   |   97.92 $\pm$ 0.04   |   87.70 $\pm$ 0.29   |   79.25 $\pm$ 1.61   |   60.24 $\pm$ 0.42   |   43.39 $\pm$ 1.15   |
> > > | Forward-VRNL   | **98.92 $\pm$ 0.06** | **98.13 $\pm$ 0.21** | **89.98 $\pm$ 0.11** | **82.35 $\pm$ 0.88** | **67.89 $\pm$ 0.30** | **53.67 $\pm$ 0.52** |
> > > | Reweight       |   98.50 $\pm$ 0.07   |   98.09 $\pm$ 0.08   |   88.55 $\pm$ 0.32   |   82.72 $\pm$ 0.38   |   60.81 $\pm$ 0.70   |   46.36 $\pm$ 0.18   |
> > > | Reweight-VRNL  | **98.77 $\pm$ 0.21** | **98.10 $\pm$ 0.16** | **89.80 $\pm$ 0.11** | **84.20 $\pm$ 0.25** | **66.62 $\pm$ 0.45** | **49.71 $\pm$ 0.64** |
> > > | VolMinNet      |   98.62 $\pm$ 0.10   |   98.03 $\pm$ 0.12   |   89.50 $\pm$ 0.18   |   83.15 $\pm$ 0.56   |   66.02 $\pm$ 0.73   |   55.17 $\pm$ 0.46   |
> > > | VolMinNet-VRNL | **98.76 $\pm$ 0.13** | **98.08 $\pm$ 0.08** | **89.64 $\pm$ 0.19** | **83.65 $\pm$ 0.32** | **66.24 $\pm$ 0.95** | **55.85 $\pm$ 0.73** |
> > >
> > > **Pair-flipping**
> > >
> > > |                |        MNIST         |        MNIST         |       CIFAR-10       |       CIFAR-10       |      CIFAR-100       |      CIFAR-100       |
> > > | -------------- | :------------------: | :------------------: | :------------------: | :------------------: | :------------------: | :------------------: |
> > > |                |    noise rate 0.2    |    noise rate 0.5    |    noise rate 0.2    |    noise rate 0.5    |    noise rate 0.2    |    noise rate 0.5    |
> > > | CE             |   98.71 $\pm$ 0.08   |   83.49 $\pm$ 3.77   |   88.63 $\pm$ 0.26   |   66.32 $\pm$ 2.44   | **61.04 $\pm$ 0.31** |   39.78 $\pm$ 0.30   |
> > > | CE-VRNL        | **98.80 $\pm$ 0.10** | **84.00 $\pm$ 3.65** | **88.71 $\pm$ 0.30** | **67.71 $\pm$ 2.04** |   61.00 $\pm$ 0.32   | **39.91 $\pm$ 0.20** |
> > > | Forward        |   98.85 $\pm$ 0.09   |   96.45 $\pm$ 4.03   | **90.88 $\pm$ 0.23** |   83.27 $\pm$ 9.47   |   62.54 $\pm$ 0.42   |   41.96 $\pm$ 1.45   |
> > > | Forward-VRNL   | **98.88 $\pm$ 0.08** | **96.55 $\pm$ 3.88** | **90.88 $\pm$ 0.29** | **83.54 $\pm$ 9.29** | **62.78 $\pm$ 0.32** | **42.29 $\pm$ 1.23** |
> > > | Reweight       |   98.64 $\pm$ 0.07   |   95.52 $\pm$ 3.58   |   89.68 $\pm$ 0.30   |   76.03 $\pm$ 5.02   |   61.35 $\pm$ 0.66   |   40.10 $\pm$ 0.46   |
> > > | Reweight-VRNL  | **98.68 $\pm$ 0.14** | **95.97 $\pm$ 3.52** | **89.83 $\pm$ 0.30** | **76.75 $\pm$ 6.15** | **61.37 $\pm$ 0.42** | **40.30 $\pm$ 0.57** |
> > > | VolMinNet      | **99.05 $\pm$ 0.05** |   99.08 $\pm$ 0.06   |   90.73 $\pm$ 0.23   |   88.47 $\pm$ 0.61   |   69.96 $\pm$ 1.18   |   61.85 $\pm$ 1.41   |
> > > | VolMinNet-VRNL |   99.02 $\pm$ 0.08   | **99.10 $\pm$ 0.08** | **90.86 $\pm$ 0.27** | **88.77 $\pm$ 0.51** | **70.18 $\pm$ 0.50** | **63.38 $\pm$ 1.72** |

---

> > > > ### Author Response · Authors · 2022-11-15
> > > > **Response to Reviewer JfaU (Continue)**
> > > >
> > > > **Q3: How much variance encouragement is theoretically or empirically unharmful for the hard but correctly-labeled examples?**
> > > >
> > > > A3: If the loss of incorrectly-labeled examples is larger than the loss of hard but correctly-labeled examples (e.g., the number of hard but correctly-labeled examples is more than the number of incorrectly-labeled examples), VRNL should not have a large negative impact on hard but correctly-labeled examples because it can still separate hard
> > > > correctly-labeled examples from incorrectly-labeled examples.
> > > >
> > > > If the loss of incorrectly-labeled examples equals the loss of hard but correctly-labeled examples (i.e., these examples are entangled), it is hard to separate hard but correctly-labeled examples from incorrectly-labeled examples. In such a case, all existing sample selection and reweighting methods would have the same problem.
> > > >
> > > > We have conducted an additional experiment to illustrate the effect of VRNL on hard but correctly-labeled examples empirically on CIFAR-10 with $50\%$ symmetry-flipping noise.
> > > > It shows that VRNL can separate hard but correctly-labeled examples from incorrectly-labeled examples. Specifically, we first extract the hard examples. Then we analyze the training loss of incorrectly-labeled examples and the training loss of hard but correctly-labeled examples. To find out hard examples, we train a ResNet-18 model on the clean dataset and sort the cross-entropy loss of all training examples. The $30\%$ examples with the largest loss are defined to be hard examples. Then we corrupt all training examples manually by using $50\%$ symmetry-flipping noise and train a new ResNet-18 model using Forward-VRNL. The noisy validation set is used to select the best model. The best model's average loss of hard but correctly-labeled training examples is 1.107 $\pm$ 0.679 while the average loss of incorrectly-labeled training examples is 2.719 $\pm$ 0.500.
> > > >
> > > > **Q4: What is the sensitivity of the method to $\alpha$? How is optimized for each setup?**
> > > >
> > > > A4: We have empirically shown the sensitivity of the method to $\alpha$ in Appendix G in the revised version. The results show that the curve of test accuracy is smooth with the increase of $\alpha$. We could conclude that the method is not sensitive to $\alpha$ in the sense of smoothness. However, it is also important to choose a proper value of $\alpha$. To select the proper $\alpha$, our method can use the noisy validation set.  Specifically, our method is mainly designed for statistically consistent methods, which can employ a noisy validation dataset to choose hyperparameters [2].  Although VRNL adds a regularizer to the statistically consistent objective with a small trade-off parameter, it has a small influence on consistency. A noisy validation dataset could work well. We have also empirically validated this by visualizing both test accuracy and validation accuracy for each training epoch. It shows that the tendency of test accuracy follows the tendency of validation accuracy. Therefore, the noisy validation set can be used to select the best $\alpha$.
> > > >
> > > > **Q5: It seems like too-early or too-high enforcement of variances can be detrimental to fitting the cleanly labeled data.**
> > > >
> > > > A5: Too-early or too-high enforcement can have an influence on fitting the cleanly labeled data. However, because the hyper-parameter $\alpha$ of our regularization can be tuned on a noisy validation set, therefore, we could avoid too-early or too-high enforcement of variances.
> > > >
> > > > **References**
> > > >
> > > > [1] Ren, Mengye, et al. "Learning to reweight examples for robust deep learning." International conference on machine learning. PMLR, 2018.
> > > >
> > > > [2] Patrini, Giorgio, et al. "Making deep neural networks robust to label noise: A loss correction approach." Proceedings of the IEEE conference on computer vision and pattern recognition. 2017.

---

> ### Author Response · Authors · 2022-11-28
> **Do you have remaining concerns**
>
> Dear Reviewer JfaU,
>
> We have provided rebuttals to address your concerns. Do you need any more clarification?
>
> Best wishes,
> Authors

---

### Author Response · Authors · 2022-11-23
**Rolling Discussion**

Dear All Reviewers,

We appreciate your constructive comments and time! We have revised the paper following your suggestions. Hope our response addresses your concern, please kindly let us know if anything is unclear.

Best Regards, Paper3687 Authors

---

### Author Response · Authors · 2022-11-30
**Discussion invitation**

Dear Reviewers JfaU, zTmG, and VFMQ,

Thanks for your comments. During the discussion with reviewer zTmG, we found that some contents of the response can be incorporated into the paper to make the paper clear. In summary, we consider that more detailed analyses of the relationship between VRNL and existing selection/reweighting methods, the experiment results of webvision, and the brief reason why validation sets can be used to select a suitable $\alpha$ can be stated in the main text to improve the paper. We consider that we can discuss the shortages of the revised paper for further improvement.

Are there any other suggestions that can make readers understand our paper more easily and clearly? Feel free to discuss with us and we will improve the paper based on the discussion results. Thanks for your attention!

Best wishes, Authors

---

> ### Author Response · Authors · 2022-12-02
> **Discussion invitation**
>
> Dear Reviewers JfaU, zTmG, and VFMQ,
>
> Thanks again for your constructive comments. We are still looking forward to your reply. If you have any concerns, feel free to discuss them with us.
>
> Best Regards,
>
> Authors

---

> > ### Author Response · Authors · 2022-12-03
> > **Discussion invitation**
> >
> > Dear Reviewers JfaU, zTmG, and VFMQ,
> >
> > Thanks again for your time and efforts in reviewing this paper. We are still looking forward to your reply.  Are there any concerns here? We can further clarify them.
> >
> > Best wishes,
> >
> > Authors

---

> > > ### Author Response · Authors · 2022-12-06
> > > **Discussion invitation**
> > >
> > > Dear Reviewers JfaU, zTmG, and VFMQ,
> > >
> > > Thanks again for your valuable comments. We have tried our best to address the concerns and we are still looking  forward to your reply. If you have any concerns, feel free to discuss  them with us.
> > >
> > > Best Regards,
> > >
> > > Authors

---

### Decision · Program_Chairs · 2023-01-20

**Decision:**

Reject

**Justification For Why Not Higher Score:**

Reviewers had some outstanding concerns, including, lack of theoretical insight about the method, moderate to low gains over baselines, sensitivity to the variance term hyperparamter. One reviewer noted that parts of the paper overstate the contributions. Simplicity of the method is a strong point of the paper and authors should provide comparisons with other methods that share this property for label noise such as [1,2,3]

[1] Normalized Loss Functions for Deep Learning with Noisy Labels, https://arxiv.org/abs/2006.13554
[2] Constrained Instance and Class Reweighting for Robust Learning under Label Noise, https://arxiv.org/abs/2111.05428
[3] Robust Bi-Tempered Logistic Loss Based on Bregman Divergences, https://arxiv.org/abs/1906.03361



**Justification For Why Not Lower Score:**

N/A

**Metareview: Summary, Strengths And Weaknesses:**

The paper proposes a method for learning with noisy data by increasing the variance of the individual example losses in a batch. It applies the method on top of three noise robustness techniques -- two based on noisy label transition matrices and on a loss reweighting method. Experimental results on MNIST, CIFAR and Clothing1M are presented to support the claims. While the reviewers appreciate the simplicity of the method, they had some outstanding concerns, including, lack of theoretical insight about the method, moderate to low gains over baselines, sensitivity to the variance term hyperparamter. One reviewer noted that parts of the paper overstate the contributions. Simplicity of the method is a strong point of the paper and authors should provide comparisons with other methods that share this property for label noise such as [1,2,3]

[1] Normalized Loss Functions for Deep Learning with Noisy Labels, https://arxiv.org/abs/2006.13554
[2] Constrained Instance and Class Reweighting for Robust Learning under Label Noise, https://arxiv.org/abs/2111.05428
[3] Robust Bi-Tempered Logistic Loss Based on Bregman Divergences, https://arxiv.org/abs/1906.03361